# Lost in Tokenization: Context as the Key to Unlocking Biomolecular Understanding in Scientific LLMs

**Kai Zhuang**[1,2,3*]**, Jiawei Zhang**[2*]**Yumou Liu**[4*]**, Hanqun Cao**[5]**, Chunbin Gu**[5]**, Mengdi Liu**[6]**,
Zhangyang Gao**[1] **, Zitong Jerry Wang**[2] **Xuanhe Zhou**[4] **, Pheng-Ann Heng**[5] **, Lijun Wu**[1]**,
Conghui He**[1,5] **, Cheng Tan**[1†]
[1]Shanghai Artificial Intelligence Laboratory [2]Westlake University [3]Shanghai Innovation Institute
[4]Shanghai Jiaotong University [5]The Chinese University of Hong Kong
[6]Institute of Computing Technology, Chinese Academy of Sciences

## Abstract

Scientific Large Language Models (Sci-LLMs) have emerged as a promising frontier for accelerating biological discovery. However, these models face a fundamental challenge when processing raw biomolecular sequences: the *tokenization dilemma*. Whether treating sequences as a specialized language, risking the loss of functional motif information, or as a separate modality, introducing formidable alignment challenges, current strategies fundamentally limit their reasoning capacity. We challenge this sequence-centric paradigm by positing that a more effective strategy is to provide Sci-LLMs with high-level structured context derived from established bioinformatics tools, thereby bypassing the need to interpret low-level noisy sequence data directly. Through a systematic comparison of leading Sci-LLMs on biological reasoning tasks, we tested three input modes: sequence-only, context-only, and a combination of both. Our findings are striking: the context-only approach consistently and substantially outperforms all other modes. Even more revealing, the inclusion of the raw sequence alongside its high-level context consistently degrades performance, indicating that raw sequences act as informational noise, even for models with specialized tokenization schemes. These results suggest that the primary strength of existing Sci-LLMs lies not in their nascent ability to interpret biomolecular syntax from scratch, but in their profound capacity for reasoning over structured, human-readable knowledge. Therefore, we argue for reframing Sci-LLMs not as sequence decoders, but as powerful reasoning engines over expert knowledge. This work lays the foundation for a new class of hybrid scientific AI agents, repositioning the developmental focus from direct sequence interpretation towards high-level knowledge synthesis.

## 1 Introduction

The convergence of artificial intelligence and the life sciences has given rise to a new class of powerful tools: Scientific Large Language Models (Sci-LLMs). Built on Transformer architectures (e.g. BERT, GPT) that have revolutionized natural language processing (Ferruz et al., 2022), these models hold immense promise for accelerating biological discovery (Karpatne et al., 2025). From predicting protein function (Brandes et al., 2023) to designing novel therapeutics (Hie et al., 2024), Sci-LLMs such as Intern-S1 (Bai et al., 2025), Evolla (Zhou et al., 2025), and NatureLM (Xia et al., 2025) are being developed to interpret the complex "language of life" encoded in DNA, RNA, and protein sequences (Taylor et al., 2022). Early efforts have demonstrated their potential, sparking visions of an AI-driven future for scientific research. This burgeoning field has largely coalesced around two primary strategies for integrating biomolecular data (Hwang et al., 2024). The first "*sequence-as-language*" approach treats sequences as a specialized form of language, extending the model's

---

[*]Equal contribution.
[†]Corresponding author.

vocabulary to include individual amino acids or nucleotides and pre-training it on vast corpora of sequence and text data. The second "*sequence-as-modality*" approach, inspired by multimodal learning, treats sequences as a distinct modality, employing a specialized encoder (e.g., a pre-trained biological foundation model like ESM (Lin et al., 2023) and Evo (Fallahpour et al., 2025)) to generate rich embeddings that are then aligned with and injected into the language model's input space, allowing LLMs to reason over high-level features of the sequence provided by the encoder, rather than the raw sequence itself (Abdine et al., 2024; Liu et al., 2024b; de Almeida et al., 2025).

While both paradigms have shown progress, they share a fundamental, yet often overlooked, vulnerability that we term the *tokenization dilemma*. In the "*sequence-as-language*" paradigm, the tokenization process is often too granular (Rao et al., 2020; Brandes et al., 2023). By breaking down sequences into their atomic components—single amino acids or nucleotides—it destroys the very structures that carry biological meaning: functional motifs, domains, and regulatory elements (Ferruz et al., 2022). The model is consequently forced into the complicated task of re-learning these fundamental "words" of biology from a stream of disconnected "letters," a process that is both inefficient and struggles with generalization. Conversely, the "*sequence-as-modality*" paradigm, while preserving structural information within its high-fidelity embeddings, introduces a formidable alignment challenge (Hu et al., 2025). The hidden space learned by a bioinformatics encoder is governed by the principles of evolution and biophysics, a world of alpha-helices and selective pressure. The hidden space of an LLM, however, is shaped by human language. Bridging this profound semantic gap between the two modalities is a non-trivial task, and imperfect alignment can introduce ambiguity or even misinterpretation, limiting the model's ability to ground its reasoning accurately in the underlying biological reality. We are, in essence, asking these models to perform a task for which they are ill-equipped: they are becoming lost in tokenization.

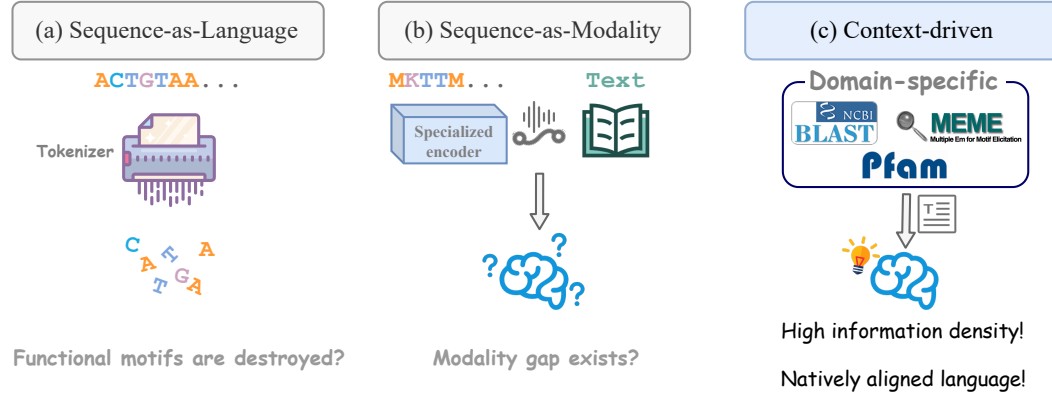

Figure 1: Paradigms for integrating biomolecular sequences into Sci-LLMs. (a) The sequence-as-language approach, tokenization fragments sequences into atomic symbols, potentially destroying functional motifs. (b) The sequence-as-modality approach preserves structure via specialized encoders but suffers from semantic misalignment with natural language. (c) The context-driven approach leverages bioinformatics tools to provide high-density, natively aligned textual context.

In this work, we challenge the prevailing sequence-centric view and propose an alternative, more effective paradigm to overcome the tokenization dilemma. We hypothesize that rather than forcing LLMs to directly decipher the noisy, low-level syntax of raw biomolecular sequences, we should leverage their core strength: reasoning over high-level, structured knowledge. Decades of accumulated biological wisdom are embedded in expert tools and databases – from BLAST for sequence homology to Pfam for conserved domains and Gene Ontology for functional terms. As shown in Figure 1, we posit that these resources can be transformed into an information-rich textual context for the LLM. This "context", presented as human-readable text, is not only information-dense, having already distilled functional insights from the raw sequence, but is also natively aligned with the LLM's linguistic domain, entirely circumventing the tokenization dilemma.

We conduct a systematic empirical study across a representative set of state-of-the-art Sci-LLMs. Surprisingly, we observe that adding the raw sequence to an already informative context often degrades performance: the sequence acts as a form of "informational noise" that confuses an otherwise

well-informed model. When both sequence and context are given, the sequence introduces misleading signals that reduce accuracy, suggesting that the true power of current Sci-LLMs lies not in their ability to serve as de novo sequence interpreters, but as sophisticated reasoning engines over integrated domain knowledge. Models that are fed high-level biological context can make insightful connections and generalizations whereas those fed only raw sequences struggle to draw any inference until they essentially "learn biology" from scratch.

## 2 RELATED WORK

### 2.1 FOUNDATION MODELS IN BIOLOGICAL REPRESENTATION

Foundation models for biological sequences have made rapid strides in representation learning. In the protein domain, large language models like ProtBERT (Elnaggar et al., 2021) and the ESM series (Lin et al., 2023; Hayes et al., 2025) are trained on massive sequence corpora, capturing signals of evolutionary conservation, structural motifs, and residue co-variation that enable downstream generalization. On the nucleotide side, models such as DNABERT (Ji et al., 2021) and the more recent Nucleotide Transformer (Dalla-Torre et al., 2025) apply $k$-mer tokenization or other subword strategies to genome-scale data, achieving high accuracy in identifying promoters, splice sites, and transcription factor binding locations. Multi-species genome models like DNABERT-2 (Zhou et al., 2024) further improve efficiency by replacing $k$-mers with Byte-Pair Encoding to accommodate longer input sequences. Meanwhile, specialized transformer architectures have extended context lengths to capture distal regulatory interactions and boost gene expression prediction (Avsec et al., 2021; Nguyen et al., 2023; 2024). Despite their powerful representational capacity, these bio-sequence foundation models largely act as "black boxes". Their internal embeddings are high-dimensional and not straightforwardly mapped to human-interpretable biological units like motifs, domains, or pathways, making it difficult to extract mechanistic insight.

### 2.2 SCIENTIFIC LARGE LANGUAGE MODELS

Large language models tailored to scientific domains (Sci–LLMs) have rapidly advanced, extending the success of general LLMs into tasks like protein or molecule design, genomic analysis, and scientific reasoning. Galactica (Taylor et al., 2022), a 120-billion-parameter model trained on a corpus of papers and knowledge bases, was introduced to store and reason over scientific knowledge. Domain-focused sequence models have also emerged: NatureLM (Xia et al., 2025), for example, is a unified sequence-based model pre-trained across proteins, nucleic acids and small molecules. Likewise, Intern–S1 (Bai et al., 2025) is a recent large multimodal MoE model (28B activated parameters) with specialized tokenization and encoders for different scientific modalities. In this work, we focus on biomolecular understanding as a representative scientific challenge: information is inherently encoded in sequences (genes or proteins), which can be expressed in textual form or as a distinct modality, making it an ideal testbed for probing how well Sci-LLMs integrate domain knowledge and whether they truly understand biological sequences.

### 2.3 EXISTING STRATEGIES IN BRIDGING SEQUENCES AND LANGUAGE

Sci-LLMs have adopted several strategies to bridge low-level biomolecular sequences with higher-level reasoning and knowledge. One common approach is treating sequences as a specialized language. Models like NatureLM (Xia et al., 2025) and Intern–S1 (Bai et al., 2025) ingest raw or tokenized sequences directly as input, training on vast datasets of sequences annotated with text so that the model learns joint representations. Another emerging strategy is treating sequences as a separate modality. For example, EvoLLaMA (Liu et al., 2024b) incorporates a protein structure encoder and a sequence encoder alongside an LLM to enable multimodal protein question-answering, and Evolla (Zhou et al., 2025) employs SaProt Su et al. (2024a) as the structure encoder. BioReason (Fallahpour et al., 2025) similarly couples a frozen DNA foundation model Evo (Nguyen et al., 2024) with a language model Qwen3 (Yang et al., 2025), so that genomic sequences are converted into contextual embeddings which the LLM can reason over in natural language. A third line of work explores agent-based or tool-augmented approaches. Rather than having a single model directly analyze sequences, the LLM is equipped with the ability to call external tools or databases as needed. Notable examples include GeneAgent (Wang et al., 2025), which self-verifies for gene-set

analysis using domain databases, and ChemCrow (Bran et al., 2023), which uses an agent to plan multi-step chemistry tasks by invoking a suite of expert tools. While all these strategies have pushed the frontier of scientific AI (Huang et al., 2025), it remains unclear how much of the success in Sci-LLMs comes from genuine reasoning over raw sequences. In this work, we adopt a deliberately context-driven baseline—providing the model with only high-level, structured annotations of the sequence. By comparing this setup to one where the model sees the raw sequence, we can assess how and when sequence information truly adds value.

## 3 PRELIMINARIES

### 3.1 THE BIOMOLECULAR UNDERSTANDING TASK

Let $\mathcal{S}$ be the space of all possible biomolecular sequences (e.g., protein, RNA, DNA, and small molecules), $\mathcal{Q}$ be the space of natural language questions about a sequence, and $\mathcal{A}$ be the space of plausible natural language answers. The general task is to learn a function $f : \mathcal{S} \times \mathcal{Q} \rightarrow \mathcal{A}$ that maps a sequence $s \in \mathcal{S}$ and a question $q \in \mathcal{Q}$ to a factually correct and relevant answer $a \in \mathcal{A}$.

A Scientific LLM, denoted as $\mathcal{M}$, aims to approximate this function by learning a set of optimal parameters $\theta$. The generation of an answer can be expressed as:

$$a = \mathcal{M}(s, q; \theta) \tag{1}$$

The fundamental distinction between the paradigms we investigate lies in how the sequence $s$ and question $q$ are represented and processed by the model $\mathcal{M}$.

### 3.2 SEQUENCE-AS-LANGUAGE

This approach, utilized by models such as NatureLM (Xia et al., 2025) and Intern-S1 (Bai et al., 2025), treats a biomolecular sequence as a specialized string of text. Let $T_{seq}$ be a tokenizer that maps a sequence $s$ into a series of tokens from a biological vocabulary, $V_{bio}$, and let $T_{text}$ be a standard tokenizer for a natural language question $q$ with vocabulary $V_{text}$. The model operates on an extended vocabulary $V_{ext} = V_{text} \cup V_{bio}$. The input to the LLM, $X_{\text{input}}$, is formed by the concatenation of the tokenized question and sequence:

$$X_{\text{input}} = [T_{\text{text}}(q); T_{\text{seq}}(s)] \tag{2}$$

The model $\mathcal{M}$ then processes this unified token sequence autoregressively to generate the answer $a$:

$$P(a|s, q) = \prod_{k=1}^{|a|} P(a_k|a_{<k}, X_{\text{input}}; \theta) \tag{3}$$

It introduces the first horn of the tokenization dilemma: the **weak representation** comes from the low-level tokenization atomizes the sequence, destroying the hierarchical structures of functional motifs. The model receives a high-dimensional but low-information-density signal, from which it must re-learn the fundamental grammar of biology, a notoriously difficult and data-intensive task.

### 3.3 SEQUENCE-AS-MODALITY

Inspired by successes in vision-language modeling, this paradigm—employed by models like Evolla (Zhou et al., 2025) and BioReason (Fallahpour et al., 2025)—treats the biomolecular sequence as a distinct, non-textual modality. A specialized, pre-trained biomolecular encoder, $\mathcal{E}_{bio} : \mathcal{S} \rightarrow \mathbb{R}^{L \times d}$, first transforms the sequence $s$ into a sequence of rich, contextualized embeddings. An alignment module, $\mathcal{A}_{\text{align}}$, then projects these biological embeddings into the LLM's semantic space, creating an aligned sequence representation $E_{\text{aligned\_seq}} \in \mathbb{R}^{K \times d}$. The final input to the LLM is a structured combination of the embedded text and the aligned sequence embeddings:

$$X_{\text{input}} = [T_{\text{text}}(q); E_{\text{aligned\_seq}}] \tag{4}$$

While this approach preserves the sequence's structural integrity, it introduces the second horn of the tokenization dilemma: the challenge of **semantic misalignment**. The semantic space of $\mathcal{E}_{bio}$ is governed by the principles of biophysics and evolution, whereas the LLM's space is structured by human linguistics and logic. The alignment module $\mathcal{A}_{\text{align}}$ must learn to bridge this profound semantic gap. Any imperfection in this translation can inject ambiguity or noise.

## 4  THE CONTEXT-DRIVEN APPROACH

In this work, we propose and investigate a third paradigm that circumvents the tokenization dilemma entirely. This approach posits that the most effective way to leverage an LLM is to provide it with what it processes best: high-quality, human-readable text.

We define a set of established bioinformatics tools as a function $\mathcal{C} : \mathcal{S} \to \mathcal{T}_{\text{context}}$, where $\mathcal{T}_{\text{context}}$ is the space of structured, human-readable textual descriptions. This function transforms a raw sequence $s$ into a high-level context $c = \mathcal{C}(s)$. The model's input deliberately omits the raw sequence $s$:

$$X_{\text{input}} = [T_{\text{text}}(q); T_{\text{text}}(c)] \tag{5}$$

The model approximates the answer's probability by conditioning only on high-level knowledge:

$$P(a|s, q) \approx P(a|c, q) = \prod_{k=1}^{|a|} P(a_k|a_{<k}, X_{\text{input}}; \theta) \tag{6}$$

This paradigm reframes the task from one of low-level sequence interpretation to one of high-level knowledge synthesis. The context $c$ is information-dense and natively aligned with the LLM's natural language space, shifting the model's role from low-level sequence interpretation to high-level knowledge synthesis and reasoning.

Specifically, we design a pipeline to generate and structure the context for any given protein sequence. First, we generate a comprehensive functional profile by executing a multi-source toolchain. InterProScan (Jones et al., 2014) is used to identify conserved domains and motifs based on the sequence's intrinsic features, while BLASTp (Altschul et al., 1990) retrieves annotations from close homologs in the Swiss-Prot database (Boeckmann et al., 2003). For novel orphan sequences lacking hits from these tools, we use the tri-modal retrieval model ProTrek (Su et al., 2024b) as a fallback to generate a basic semantic description. The outputs from these tools are then integrated into a final context using an empirically-driven hierarchical strategy. The details are in the Appendix A.

```
Structured Prompt for Context-Driven Reasoning

You are a senior systems biologist.  Analyze the input
information to answer the given question.
---------
Question:[User's Question Text]
---------
Conserved Domains (from Pfam):
[FOR EACH Pfam entry IN Pfam]:
- {the description of detected conserved domains/motifs}
Functional Annotations (from Homology via BLASTp):
- GO terms associated with the homolog:
- {the GO terms of the homolog}
Fallback Semantic Analysis (from ProTrek):
[ONLY if no homology or domain data is available]
[FOR EACH ProTrek entry In Protrek]:
- {the description of Protrek}
-------
Answer:{answer}
```

A central concern in fair evaluation is the prevention of information leakage. Our context-driven approach is explicitly designed to avoid label leakage along two complementary axes:

**Intrinsic analysis rather than identity lookup.** We employ InterProScan to detect conserved domains and motifs intrinsic to the query sequence. This constitutes an *ab initio*, feature-based analysis grounded in domain knowledge bases, not in annotation records of the query protein itself. Consequently, even for genuinely novel proteins, recognizable elements such as a kinase domain can be identified without ground-truth labels.

**Homology-based inference rather than direct annotation matching.** When using BLASTp, we restrict our context-driven approach to reading GO annotations from the homologous sequences, rather than from the query protein's own record. This reflects standard bioinformatics practice: predicting the function of unknown sequences by analogy to characterized homologs rather than simply retrieving pre-annotated answers.

## 5 THE TOKENIZATION DILEMMA IN PRACTICE

### 5.1 THE PRIMACY OF CONTEXT OVER SEQUENCE

Following a standardized protocol inspired by Evolla (Zhou et al., 2025), our benchmark focuses on three fundamental aspects of protein biology: molecular function, metabolic pathway involvement, and subcellular localization. For each protein in our test set, we generated queries corresponding to these categories (e.g., "What is the function of this protein?"). To ensure a set of factually grounded and verifiable ground truths, a question was only included if its corresponding annotation field was explicitly present in the source database entry, from which the answer was directly excerpted. Performance was quantified using an automated pipeline, leveraging a general-purpose LLM as an expert judge, a metric we term the LLM-Score. A detailed description of the dataset construction, evaluation protocol, and prompt design is provided in Appendices B and C. We evaluate the performance of both specialized Sci-LLMs and leading general-purpose LLMs across three distinct input configurations: (i) Sequence-Only, where the model receives only the raw protein sequence; (ii) Sequence + Context, a combined input; (iii) Context-Only, where the model receives only the high-level context. The results are presented in Table 1.

Table 1: Comparison of performance across specialized Sci-LLMs and general-purpose LLMs on our protein QA benchmark. ✓ indicates that the corresponding input modality was provided to the model. Results are reported on three task-specific subsets—*Function* (Func.), *Pathway* (Path.), and *Subcellular Location* (Sub. Loc.)—as well as the overall average (All). The best score for each model is underlined, and the overall best performance across all models is highlighted in bold.

| Model | Sequence | Context | Func. | Path. | Sub. Loc. | All |
|---|---|---|---|---|---|---|
| ***Specialized Sci-LLMs*** | | | | | | |
| Intern-S1 | ✓ | | 20.57 | 26.56 | 69.75 | 43.33 |
| Intern-S1 | ✓ | ✓ | 74.18 | 98.85 | 93.00 | 84.03 |
| Intern-S1 | | ✓ | 76.22 | 97.60 | 95.60 | 86.15 |
| Evolla | ✓ | | 40.23 | 72.71 | 79.76 | 59.93 |
| Evolla | ✓ | ✓ | 57.46 | 84.69 | 83.05 | 70.53 |
| Evolla | | ✓ | 65.77 | 83.33 | 81.88 | 74.02 |
| NatureLM | ✓ | | 3.58 | 5.52 | 10.45 | 6.82 |
| NatureLM | ✓ | ✓ | 42.33 | 64.25 | 32.30 | 38.86 |
| NatureLM | | ✓ | 44.77 | 51.35 | 32.51 | 39.50 |
| ***General LLMs*** | | | | | | |
| Deepseek-v3 | ✓ | | 10.98 | 24.54 | 74.72 | 40.77 |
| Deepseek-v3 | ✓ | ✓ | 77.40 | 91.35 | 94.75 | 86.03 |
| Deepseek-v3 | | ✓ | 75.79 | 93.96 | 93.65 | 84.99 |
| Gemini2.5 Pro | ✓ | | 10.40 | 13.85 | 77.58 | 41.25 |
| Gemini2.5 Pro | ✓ | ✓ | 79.12 | 94.17 | 94.65 | 86.98 |
| Gemini2.5 Pro | | ✓ | 79.17 | 98.65 | 94.56 | **87.19** |
| GPT-5 | ✓ | | 19.64 | 17.08 | 64.15 | 39.83 |
| GPT-5 | ✓ | ✓ | 79.89 | 89.48 | 71.30 | 76.45 |
| GPT-5 | | ✓ | 77.25 | 85.73 | 73.05 | 75.76 |
| Qwen3-235B-A22B | ✓ | | 13.67 | 19.90 | 37.17 | 39.51 |
| Qwen3-235B-A22B | ✓ | ✓ | 76.62 | 96.35 | 94.78 | 85.90 |
| Qwen3-235B-A22B | | ✓ | 75.63 | 92.19 | 94.28 | 84.99 |

> **Takeaway**: Raw biomolecular sequences, when provided alone, offer limited utility and, when combined with context, consistently act as informational noise.

Our findings demonstrate that the Context-Only approach is dramatically superior, confirming our hypothesis: *LLMs excel when they can leverage their core strength of reasoning over structured*

*knowledge.* Even more revealing is the consistent performance degradation observed in the Sequence + Context configuration. The inclusion of the raw sequence alongside its high-level summary resulted in a lower score. For instance, Evolla's score dropped from 74.02 to 70.53, and Intern-S1's from 86.15 to 84.03. This counter-intuitive result provides evidence that raw sequences, in their current tokenized form, are not merely redundant but actively detrimental, acting as a source of noise. The models become, as we posited, "lost in tokenization". This phenomenon underscores the profound limitations of existing sequence tokenization paradigms.

## 5.2 DECONSTRUCTING THE DILEMMA I: THE WEAK REPRESENTATION

We visualize the embeddings of the outputs, where ground-truth classes were established by clustering homologous proteins using MMseqs2 at a 50% sequence identity threshold. For each model, we extracted the final-layer embeddings for their outputs. We employed t-SNE (Maaten & Hinton, 2008) to project them into a 2D space. The quality of the resulting functional separation was then quantified by performing clustering on the high-dimensional embeddings and calculating the Adjusted Rand Index (ARI) against the MMseqs2 ground-truth clusters. For our context-driven approach, we generated embeddings from the structured context itself using the text embedding model Qwen-embedding (Zhang et al., 2025). The results are visualized in Figure 2.

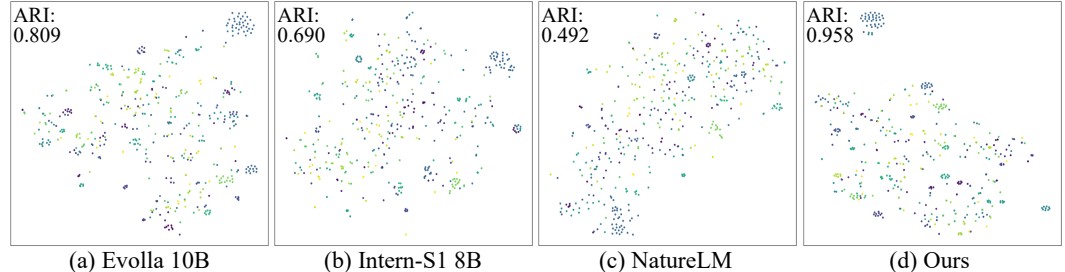

| (a) Evolla 10B | (b) Intern-S1 8B | (c) NatureLM | (d) Ours |

Figure 2: The visualization of representation spaces.

> **Takeaway**: Simple context provides a vastly superior functional representation of proteins compared to both sequence-to-language/modality strategies.

The visualizations confirm the weak representation horn of the tokenization dilemma. The sequence-as-language models, NatureLM (c) and Intern-S1 (b), exhibit highly disorganized latent spaces, quantitatively confirmed by their low ARI scores of 0.492 and 0.690, respectively. Evolla (a), which employs the sequence-as-modality paradigm, demonstrates a significantly more structured representation, highlighting the benefit of using a specialized sequence encoder. However, both paradigms are dramatically outperformed by our context-driven approach (d). The representation derived purely from the textual context achieves near-perfect functional separation.

## 5.3 DECONSTRUCTING THE DILEMMA II: THE SEMANTIC MISALIGNMENT

While the sequence-as-modality paradigm, exemplified by Evolla, overcomes the weak representation problem, it introduces a more subtle yet equally critical challenge: semantic misalignment. The specialized encoder and the generalist LLM operate in fundamentally different semantic worlds—one governed by biophysics, the other by linguistics. We performed a layer-wise representational analysis of the Evolla-10B model, tracing the informational journey of a protein sequence from its biological embedding to its final interpretation by the language model. As shown in Figure 3, the initial SaProt encoder generates a well-structured latent space. As the Q-Former works to translate these biological embeddings for the LLM, the functional clarity begins to blur.

> **Takeaway**: The degradation of functional representation stems not from the initial protein encoding, but from the subsequent semantic alignment to the language model.

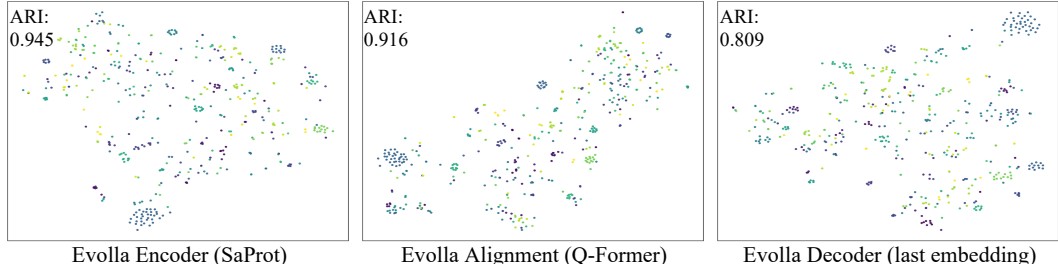

Figure 3: Visualization of representation spaces at different stages within the Evolla-10B model.

## 5.4 DEGRADING PHENOMENON ACROSS TIME

We curated a dataset by randomly sampling about 100 proteins for each year from 1995 to 2024 based on the first publication year. The relationship between a protein's first publication year and the models' LLM-Scores is visualized in Figure 4.

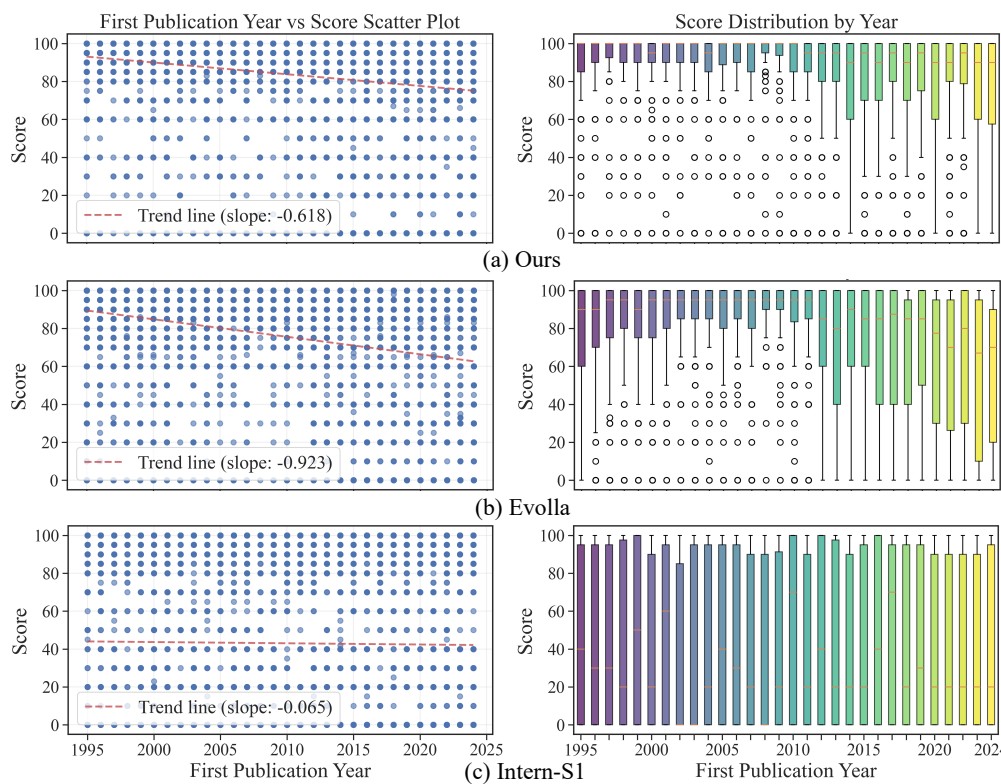

Figure 4: Analysis of model performance versus protein's first publication year.

For this analysis, our context-driven approach employed DeepSeek-V3 (Liu et al., 2024a) as its base LLM to ensure a fair comparison against models with similar training data cut-off dates. **Our context-driven approach (a)**, while maintaining the highest overall performance, exhibits a slight negative trend over time due to the diminishing availability of rich, homologous information in the knowledge bases. For very recent proteins, homology-based tools like BLAST find fewer well-characterized relatives, leading to a sparser context and thus slightly less precise answers. **The sequence-as-modality model, Evolla (b)**, displays a much more pronounced degradation. Its performance on well-studied proteins from the 1990s and early 2000s is strong, but it deteriorates significantly for proteins discovered in the last decade. It is crucial to note that Evolla's training data, sourced from Swiss-Prot Release (202303), has a temporal bias. Therefore, part of this decline can be attributed to its lack of exposure to the most recent protein data. However, this training bias alone does not fully account for the steepness of the collapse. The trend suggests a deeper issue: Evolla's encoder appears to rely heavily on the dense web of evolutionary information available for older, larger protein families. When faced with recent, potentially more unique proteins that lack this

deep evolutionary context—a problem exacerbated by its training data cutoff—the encoder's ability to generate meaningful representations weakens considerably. **The sequence-as-language model, Intern-S1 (c)**, shows a performance profile that is almost entirely flat and consistently low across the entire 30-year period. This lack of temporal trend, combined with its overall poor performance, indicates a fundamental failure to extract meaningful biological signals from the raw sequence.

> **Takeaway**: Our context-driven approach demonstrates superior generalization: (i) *Robustness to sequence novelty*: Unlike Sci-LLMs which suffer collapsing on proteins dissimilar to training data, our context maintains high accuracy regardless of sequence identity. (ii) *Temporal stability*: Our approach's performance degrades far more gracefully over time on recently discovered proteins compared to other paradigms.

The above dual robustness confirms that reasoning over stable, high-level knowledge is a more robust foundation for AI in biology than relying on the difficult task of raw sequence interpretation.

## 5.5 QUANTITATIVE ANALYSIS OF REAL-WORLD EFFICIENCY

We conducted a detailed, three-way analysis to quantify the trade-offs between computational cost, time, and performance. We compare: (i) A general LLM baseline Deepseek-v3, which feeds the raw sequence to the DeepSeek-V3 API and yields a performance score of **40.77**, (ii) A specialized Sci-LLM Evolla, which requires a high-end GPU and achieves a performance score of **59.93**, and (iii) our context-driven method, which uses bioinformatics tools on a CPU plus the DeepSeek-V3 API to achieve a state-of-the-art performance score of **84.99**. We conducted the analysis across both single-sequence and batch-processing scenarios. The results, based on AWS on-demand pricing, are summarized in Table 2, with the cost estimation detailed in Appendix M.

For individual queries, our method is not only dramatically more effective but also approximately **23 times cheaper** and **1.3 times faster** than the specialized end-to-end model. The true efficiency of our pipeline is most evident in high-throughput research. In a realistic, large-scale scenario, our method is nearly **30 times cheaper** and, critically, **154 times faster** than Evolla on a per-sequence basis. While the direct API baseline is cheaper, its poor performance renders it unsuitable for scientific applications. In summary, the small cost of running established bioinformatics tools is a highly effective investment, unlocking a $> 2\times$ performance gain over the LLM baseline while simultaneously outperforming and costing less than specialized end-to-end models, especially at scale.

Table 2: Comparative analysis of inference efficiency.

| Method | Mode | Input to LLM | Avg. Time | Avg. Cost |
|---|---|---|---|---|
| Deepseek-v3 | Single | Raw sequence | ~30s | $0.0005 |
| Evolla | Single | Raw sequence | ~90s | $0.0690 |
| Our Method | Single | Context | ~70s | $0.0030 |
| Evolla | Batch | Raw sequence | ~20s | $0.0152 |
| Our Method | Batch | Context | ~0.13s | $0.0005 |

## 5.6 WET-LAB VALIDATION ON NOVEL SEQUENCES

We curated a set of novel functional protein sequences obtained from wet-lab experiments. They were unpublished at the time of our analysis and absent from major databases, including Swiss-Prot, thereby representing a true test of performance on unseen data. The task was formulated as a binary classification problem for two distinct protein families: Rhodopsin and PETase. For each sequence, the LLM was prompted to predict its classification. As shown in Figure 5, our method achieves 100% accuracy for Rhodopsin and 97.3% accuracy for PETase. While Evolla (Figure 6) attains a reasonable 80.0% accuracy on Rhodopsin, it fails catastrophically on PETase. It may caused by its training data bias. A detailed description of the wet-lab validation is shown in Appendix N.

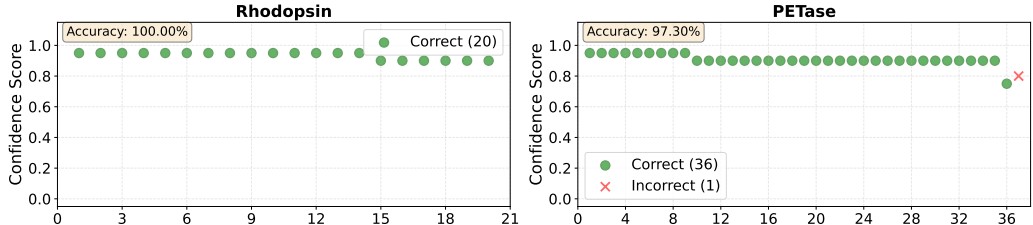

Figure 5: Sample-level performance of our context-driven method + Deepseek-v3.

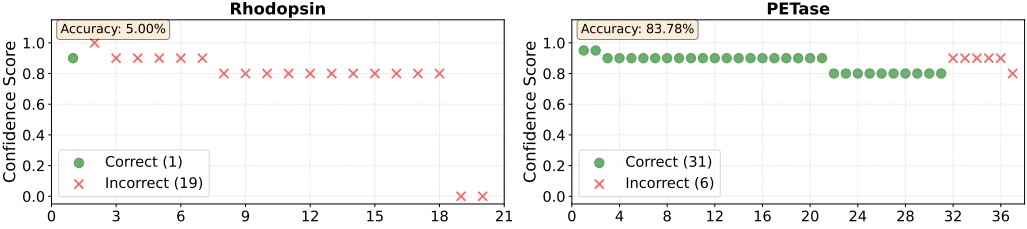

Figure 6: Sample-level performance of Evolla.

## 6 CONCLUSION AND LIMITATION

In this work, we confronted a fundamental challenge at the heart of modern Sci-LLMs: the tokenization dilemma. We demonstrated that current paradigms, whether treating biomolecular sequences as a specialized language or as a distinct modality, are fundamentally handicapped by issues of weak representation and semantic misalignment. Our central contribution is the validation of a third paradigm that resolves this dilemma. By shifting the focus from low-level sequence interpretation to high-level knowledge synthesis, our context-driven approach entirely circumvents the tokenization problem, as illustrated in the conceptual landscape of Figure 7. Notably, our approach is also computationally efficient, as it leverages generalist LLMs without the costly retraining required by domain-specific Sci-LLMs.

While our findings are compelling, we acknowledge several limitations. For truly novel orphan proteins from unexplored regions of the protein universe, our method's performance may be constrained. Furthermore, our current analysis has primarily focused on proteins; although we provide some preliminary exploration in Appendix G, a more comprehensive treatment remains open for future research.

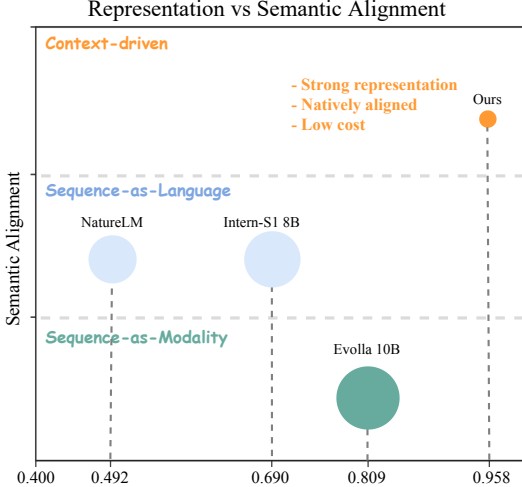

Figure 7: The trade-off landscape of representation vs. semantic alignment. The x-axis quantifies the quality of the biological representation (measured by ARI), while the y-axis conceptually represents the degree of semantic alignment with natural language. The area of each circle is proportional to the computational cost, with larger circles indicating higher computational expenses.

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

CONTENTS

## A    CONTEXT DETAILS

Our context-driven consists of three stages: (1) generation of a multi-source evidence profile, (2) hierarchical construction of a textual context, and (3) context-based inference by LLMs.

### A.1    CONTEXT GENERATION

For any given biomolecular sequence (we use proteins as the running example), we first generate a comprehensive functional profile using a deliberately crafted, multi-source toolchain.

**Input**: A single protein FASTA sequence.

**Toolbox Execution**:

1. **Feature-intrinsic domain analysis:** We first scan the input sequence with **Inter-ProScan** (Jones et al., 2014) to identify conserved domains and key motifs by integrating multiple signature libraries (e.g., Pfam, PROSITE, SMART). This step is an analysis grounded in *intrinsic* sequence features: even for a completely novel, unannotated protein, InterProScan can recognize known modular features. We extract textual descriptions of detected domains together with any directly linked Gene Ontology (GO) annotations (Ashburner et al., 2000) .

2. **Homology-based functional inference.** In parallel, we run **BLASTp** (Altschul et al., 1990) against a curated reference database (e.g., Swiss-Prot) to retrieve close homologs. We *transfer* GO annotations from the most similar sequences to the query. *Critically, we never use the query's own (possibly unknown) labels.*

3. **Information integration.** We merge GO evidence obtained from InterProScan (feature-intrinsic) and from BLASTp (homology-based) to form a comprehensive functional profile.

4. **Fallback mechanism.** In rare cases where neither BLASTp nor InterProScan yields informative signals ("orphan" sequences), we invoke **ProTrek** (Su et al., 2024b) to synthesize a concise, model-based textual description that serves as minimal context.

## A.2 Context Construction

The raw outputs from these tools can be redundant, conflicting, or noisy, especially for novel proteins. A naive combination of all outputs is suboptimal. Therefore, based on rigorous empirical evaluation (see Ablation Study), we developed a hierarchical strategy designed to gracefully handle the spectrum of protein novelty:

1. **Prioritizing High-Confidence Homology:** Our analysis revealed that for generating a list of candidate GO terms, the single most reliable source is the annotation of the top homolog found by BLAST. This strategy maximizes precision while maintaining high recall (see Table 3).

2. **Integrating Domain Information:** Pfam motifs identified by InterProScan are added as a separate, complementary source of evidence, providing structural and functional context.

3. **Semantic Evidence:** Our experiments showed that ProTrek's semantic hits, while powerful, could introduce noise when combined with high-quality GO/Pfam data. Therefore, ProTrek's output is used as a fallback—it is only added to the context when primary sources like GO and Pfam are sparse or absent.

This empirically-driven, hierarchical process culminates in a final textual context engineered to be as factually dense and noise-free as possible, ready for the final inference stage.

## A.3 Context-based Inference

The final stage transforms the structured biological evidence into a query that the LLM can process. The constructed context and the original user question are formatted into a unified prompt using a predefined template as shown in Figure 4. The LLM's role is to act as a knowledge synthesizer. It processes the prompt, which contains a series of factual statements derived from the context. This final step leverages the LLM's core strength in natural language understanding and reasoning, entirely bypassing the need for it to interpret the complex, low-level syntax of the raw biomolecular sequence itself.

## B Dataset details

### B.1 Protein Dataset

To ensure a comprehensive and fair evaluation of our model against Evolla, we employed a multi-faceted dataset strategy. This approach incorporates not only the benchmark datasets used in the original Evolla study but also a dataset we have meticulously reconstructed to address specific limitations in their evaluation methodology. Our assessment is primarily based on the following datasets:

1. **Our Benchmark Dataset**
   We reconstructed part of the Evolla dataset to create a more rigorous and objective benchmark.

This subset, containing proteins with less than 30% sequence identity to the training set, is critical for assessing a model's generalization capabilities on novel proteins. Our reconstruction was guided by the following principles:

- **Standardized Questions:** We replaced open-ended queries with a fixed set of three targeted question templates: "What is the function of this protein?", "What is the pathway of this protein?", and "What is the subcellular location of this protein?"
- **Conditional Question Generation:** To ensure every question has a verifiable answer, we only generated a specific question if the corresponding field ("Function," "Pathway," or "Subcellular location") was explicitly present in the protein's source database entry.
- **Authentic Ground Truth:** Crucially, our ground truth answers are direct excerpts from the protein's database entry. Unlike the original dataset, we did not use an LLM to generate answers, thereby ensuring the objectivity and factual accuracy of the GT and creating a more reliable scoring standard.

The results for our benchmark dataset are presented in Table 1, where we compare the performance of different methods using sequence-only, context-only, and sequence+context inputs.

2. **Enzyme Commission (EC) Number Dataset**
   To evaluate the model's performance on a specific, structured bioinformatics task, we also utilized the EC Number dataset from the Evolla study. This task requires the model to accurately predict the functional class of enzymes, which is a standardized and important functional annotation task. Testing on this dataset allows us to gauge the model's capabilities in handling classification-oriented protein function prediction, and is presented in Figure 8.

3. **Time-Split Dataset**
   To investigate the model's performance over time and the impact of sequence novelty, we curated a dataset by randomly sampling about 100 proteins for each year from 1995 to 2024 based on the first publication year. This allows us to analyze the relationship between a protein's first publication year and the model's LLM-Score, as shown in the "Degrading Phenomenon Across Time" section (Figure 4). The time-split dataset is valuable for understanding how well the model generalizes to older versus more recent proteins, and whether its performance degrades as the data becomes more novel, less represented in training, or more temporally distant from the model's training data cutoff.

4. **Mol-Instruction Dataset**
   To further evaluate the model's ability to handle diverse protein tasks, we tested it on the Mol-Instruction benchmark (Fang et al., 2024). This benchmark provides datasets for assessing molecular understanding across a range of tasks. We specifically evaluated the model on three functionally distinct protein subsets: Catalytic Activity, General Function, and Protein Function. The performance on these datasets, as illustrated in the "Performance on Protein Function Prediction" section (Figure 10), highlights the model's ability to accurately predict protein function even across different protein families, demonstrating the robustness and flexibility of our context-driven approach.

## B.2 DNA DATASET

To assess whether our context-driven paradigm extends beyond proteomics, we evaluated its performance on a DNA-based mechanistic reasoning task. We utilized the KEGG Disease Pathway dataset curated by BioReason (Fallahpour et al., 2025), which provides a unique benchmark for connecting genomic variants to disease phenotypes through multi-step biological pathways. Each entry in the dataset consists of a reference and a variant DNA sequence, the associated KEGG pathway definition, and the ground-truth disease outcome. The task requires the model to reason from the mutation and its functional context to predict the correct disease.

We designed three experimental setups to investigate the impact of different data configurations on the model's performance. The first setup included only pathway-related contextual information (context-only). The second setup incorporated both the pathway context and the raw DNA sequence data (context and sequence). The third setup focused solely on the DNA sequence itself (sequence-only). These configurations allowed us to evaluate the effect of using context, sequence, or both on the model's ability to predict mutations.

The KEGG dataset's comprehensive pathway data, paired with precise mutation annotations, provided a solid foundation for designing these experiments. By varying the inclusion of sequence and context information, we aimed to assess the model's ability to predict the effects of DNA mutations based on both pathway context and raw sequence data.

## C  EVALUATION METRIC

To conduct a comprehensive and multi-dimensional assessment of our model's performance, we designed specific evaluation metrics tailored to each of our distinct tasks.

### C.1  LLM-SCORE FOR GENERAL PROTEIN QA TASKS

For the open-ended protein question–answering task, traditional metrics based on lexical overlap (e.g., BLEU, ROUGE) are inadequate for assessing the semantic accuracy and factual consistency of generated answers. To address this, we adopted an automated evaluation methodology leveraging a LLM as an adjudicator, which we term the LLM-Score. The core principle of this metric is to use a powerful, independent third-party LLM (in this case, a DeepSeek-V3 (Liu et al., 2024a) model) to score the quality of our model's generated answer against the ground truth. The evaluation process is as follows:

1. **Prompt Construction:** We embed the generated answer and the ground truth answer into a carefully designed prompt template. This prompt instructs the adjudicator LLM to act as an expert biologist and perform a holistic evaluation based on factual accuracy. The exact prompt is shown below.

> **LLM-Score Adjudicator Prompt**
>
> ```
> As an expert biologist, you are assigned to check one paragraph
> is aligned with facts or not.  You will receive some facts, and
> one paragraph.  Score the paragraph between 0 to 100.
> The score should be the format of {"score":  score}
> ---------
> ```
> **Here's the facts:**
> ```
> [Ground Truth Text from Database]
> ---------
> ```
> **Here's the paragraph:**
> ```
> [Generated Answer from Model to be Scored]
> ```

2. **Score Generation:** The adjudicator LLM processes the prompt and returns a numerical score on a scale from 0 to 100, where a higher score indicates that the generated answer is of higher quality and more closely aligned with the ground truth.

3. **Score Extraction and Aggregation:** A robust parsing function extracts the numerical score from the LLM's textual response. The model's final performance on the dataset is reported as the *average LLM-Score* across all test samples.

This approach moves beyond surface-level text matching to provide a deeper, more semantically aware assessment of the model's ability to comprehend and articulate biological knowledge.

### C.2  HIERARCHICAL METRICS FOR EC NUMBER PREDICTION

The Enzyme Commission (EC) number is a four-level hierarchical classification system (e.g., `A.B.C.D`). A proficient model should be rewarded not only for predicting the exact four-digit code but also for correctly identifying the broader functional classes at higher levels of the hierarchy. Therefore, an exact-match accuracy metric at a single level is insufficient.

To capture this, we implemented a more nuanced, hierarchical evaluation scheme. We calculate **F1-Score** at each of the four functional levels.

The methodology is as follows:

- **Hierarchical Matching:** To evaluate performance at `level-N`, all predicted and ground truth EC numbers are truncated to their first N digits for comparison. For example, at `level-3`, a prediction of `1.2.3.5` is considered a match for a ground truth of `1.2.3.4`.

- **Multi-Label Formulation:** As a single protein can be associated with multiple EC numbers, the task is treated as a multi-label classification problem.

- **Micro-Averaging:** We compute the total number of True Positives (TP), False Positives (FP), and False Negatives (FN) by aggregating their counts over the entire test set. Global Precision, Recall, and F1-Score are then calculated from these aggregate sums.

$$\text{Precision}_{\text{micro}} = \frac{\sum \text{TP}_i}{\sum \text{TP}_i + \sum \text{FP}_i} \tag{7}$$

$$\text{Recall}_{\text{micro}} = \frac{\sum \text{TP}_i}{\sum \text{TP}_i + \sum \text{FN}_i} \tag{8}$$

$$\text{F1}_{\text{micro}} = 2 \cdot \frac{\text{Precision}_{\text{micro}} \cdot \text{Recall}_{\text{micro}}}{\text{Precision}_{\text{micro}} + \text{Recall}_{\text{micro}}} \tag{9}$$

This suite of metrics provides a comprehensive view of the model's predictive accuracy at varying degrees of granularity and effectively handles the multi-label nature of the data, offering a more equitable measure of true performance.

## D    QUANTITATIVE BENCHMARK: EC NUMBER PREDICTION

To further validate our central thesis on the primacy of context over sequence, we introduce a quantitative benchmark: EC number prediction. This task provides an objective, verifiable measure of a model's ability to understand a protein's precise biochemical function. The hierarchical nature of EC numbers allows us to evaluate performance at four increasing levels of specificity (from 1-digit to 4-digit matches), with the F1-Score serving as our primary metric.

We compare two categories of models: "Sequence-Only" models, which include both general-purpose LLMs and CLEAN (Yu et al., 2023) (a model specifically trained for this task), and "Context-Driven" models, which leverage the contextual information as described in the main text. The comparative performance is visualized in Figure 8.

The results presented in Figure 8 are unequivocal and offer several key insights:

- **Failure of Sequence-Only LLMs:** In the Sequence-Only setting, both general-purpose LLM DeepSeek-V3 and Sci-LLM Intern-S1 perform poorly. Their F1-scores plummet as the required specificity increases, demonstrating their inability to decipher complex enzymatic function from raw sequence data alone. This reinforces our "lost in tokenization" hypothesis.

- **Context Outperforms Specialization:** CLEAN, a model specifically trained on sequences for EC prediction, establishes a respectable baseline. However, every model in the Context-Driven category outperforms CLEAN across the first three levels of precision (1-digit, 2-digit, and 3-digit). This demonstrates that providing high-level context to a general model is more effective than training a specialized model on sequence data for these levels.

- **Robustness of the Context-Driven Approach:** While all models show a natural decline in performance as the task becomes harder (from 1-digit to 4-digit prediction), the context-driven models exhibit a much more graceful degradation. Gemini-2.5 Pro, using only context, achieves the highest F1-score of 0.406 on the most challenging 4-digit prediction task, a score comparable to the specialized CLEAN model's performance (0.408).

In summary, this quantitative benchmark provides strong, direct evidence that high-level biological context is a far more effective and reliable representation of protein function for LLMs than the raw amino acid sequence. It enables general-purpose models to excel at highly specific bioinformatics tasks without needing task-specific architectures or fine-tuning.

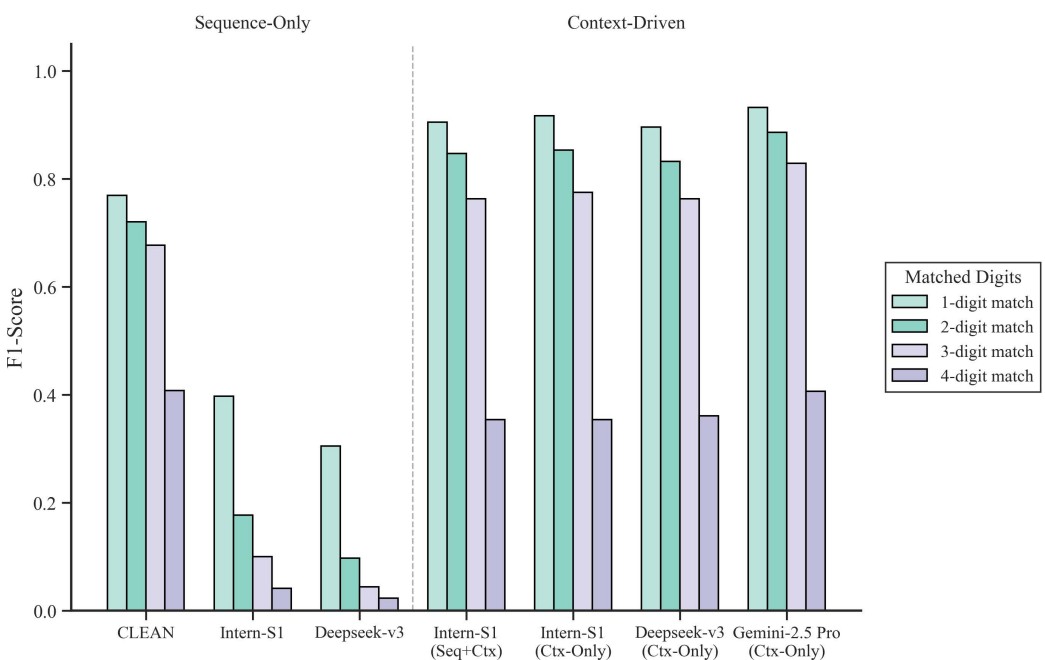

Figure 8: **Performance on EC Number Prediction (F1-Score).** The plot is divided into Sequence-Only models (left) and Context-Driven models (right). A clear and dramatic performance gap is visible between the two groups. Context-driven approaches significantly outperform even specialized sequence-based models like CLEAN, especially at higher levels (3 and 4 digits).

# E  ABLATION STUDY

To dissect the contribution of each component within our framework, we conducted a comprehensive ablation study. The experiments were performed on our benchmark dataset, which is ideal for evaluating generalization, as it contains proteins with less than 40% sequence identity to the Evolla training set. We systematically evaluated the performance by providing different combinations of contextual information—Pfam, GO, and ProTrek—to the DeepSeek model. The results, summarized in Table 3, reveal the individual and synergistic effects of these components.

Table 3: Ablation study of context components on our benchmark dataset. Scores reflect the model's performance when provided with different combinations of contextual information. Our final, conditional approach yields the best result.

| Context Components Provided | LLM Score |
|---|---|
| ***Single Components*** | |
| Pfam only | 74.90 |
| GO only | 84.02 |
| ProTrek only | 66.44 |
| ***Pairwise Combinations*** | |
| Pfam + GO | 84.60 |
| Pfam + ProTrek | 77.00 |
| GO + ProTrek | 77.78 |
| ***Full Combinations*** | |
| Pfam + GO + ProTrek (Unconditional) | 81.56 |
| **Pfam + GO + ProTrek (Conditional)** | **84.99** |

**Analysis of Individual Components**    The results from single-component experiments establish a clear hierarchy of information value. Gene Ontology (GO) annotations emerge as the most powerful single source of context, achieving a high score of 84.02 on its own. Pfam provides a moderately strong signal, scoring 74.90. In contrast, ProTrek alone is the least informative component, with a score of 66.44, suggesting its raw output may be noisy or less directly useful for functional queries.

**Synergistic and Antagonistic Effects in Combinations**    Combining Pfam and GO yields a score of 84.60, a slight improvement over GO alone, indicating a positive, synergistic relationship where Pfam provides complementary information. However, a critical observation arises when combining components with ProTrek. Both 'Pfam + ProTrek' (77.00) and 'GO + ProTrek' (77.78) perform worse than their stronger counterparts (Pfam and GO, respectively) alone. This trend is amplified when all three are combined unconditionally ('Pfam + GO + ProTrek'), resulting in a score of 81.56, which is substantially lower than 'Pfam + GO'. This strongly suggests that naively adding ProTrek's information introduces noise that dilutes the high-quality signals from Pfam and GO, ultimately degrading the model's performance.

**Justification for the Conditional Strategy**    Based on this insight, we implemented our final, conditional strategy: ProTrek information is only included as a fallback when both Pfam and GO annotations are unavailable for a given protein. This intelligent inclusion prevents ProTrek from interfering with higher-quality data while still providing a baseline of information for sparsely annotated proteins. As shown in the final row of Table 3, this conditional approach achieves the highest score of 84.99. It effectively captures the synergy of Pfam and GO while mitigating the negative, noisy impact of ProTrek, thus justifying its selection as our final methodology.

## F    Impact of Semantic Alignment on Mutation Sensitivity

A critical capability for any protein model is the ability to detect and represent the effects of small sequence variations, such as point mutations. To investigate how the internal mechanisms of sequence-as-modality models like Evolla handle such changes, we conducted an analysis on the feature representations before and after its Q-Former alignment module.

We introduced a series of mutations (from 1 to 4 differing sites) into a sample protein sequence. We then extracted the resulting feature embeddings at two key stages: (1) directly from the SaProt protein encoder, and (2) after they had been processed by the Q-Former. The difference between the pre-mutation (wild-type) and post-mutation embeddings at each stage was then visualized and quantified. The results are presented in Figure 9.

**High Sensitivity at the Protein Encoder Stage**    As expected, the SaProt protein encoder is highly sensitive to sequence mutations. The visualizations (Figure 9, top row) show clear, localized changes in the feature map corresponding to the mutation sites. Quantitatively, while the cosine similarities between pre- and post-mutation embeddings remain high (0.980–0.995), the Euclidean distances are substantial (ranging from 13.3 to 26.2). This confirms that the encoder accurately captures the perturbation caused by the mutation, altering the feature vector in a meaningful way. This sensitivity is the foundation required for any downstream mutation effect analysis.

**Loss of Sensitivity After Q-Former Alignment**    A starkly different picture emerges after the features pass through the Q-Former. The difference heatmaps (Figure 9, bottom row) are almost entirely uniform, indicating a negligible change between the wild-type and mutated representations. This visual observation is confirmed by the quantitative metrics: the cosine similarities are nearly perfect (approaching 1.0, e.g., $> 0.9999$), and the Euclidean distances (5.8–9.9) are significantly smaller than those observed from the encoder.

**Implications for Downstream Tasks**    This analysis reveals a critical limitation of the sequence-as-modality paradigm employed by models like Evolla. The Q-Former, in its role of compressing and aligning the detailed protein features into a fixed set of tokens for the language model, effectively "smooths out" or discards the fine-grained information related to single point mutations. While this may be sufficient for generating high-level functional descriptions, it renders the final representation insensitive to the subtle yet critical differences that underpin tasks like mutation effect prediction,

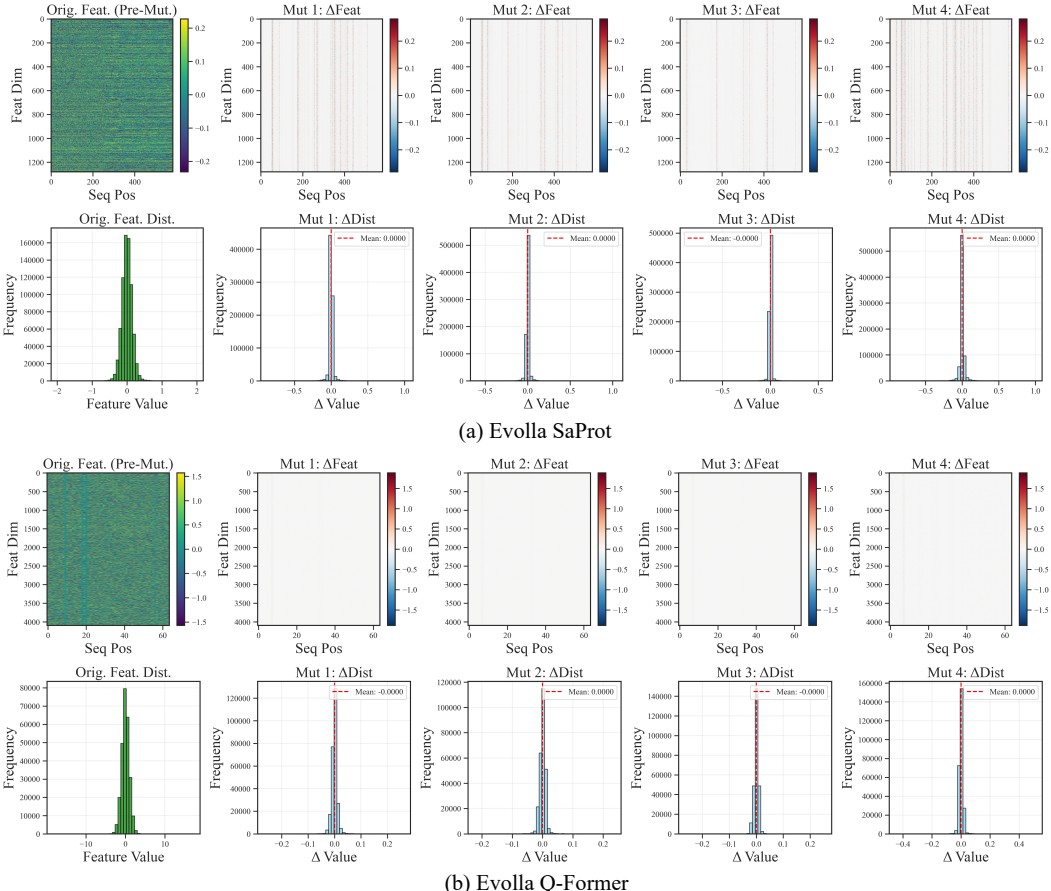

Figure 9: Effect of mutations on internal representations of Evolla. The top row shows feature differences from the **SaProt encoder**, and the bottom row from the **Q-Former**. Heatmaps visualize the difference vector ('mutated - original'). SaProt's representation is clearly perturbed by mutations, showing localized and significant changes. In contrast, the Q-Former's output shows almost no change, indicating that the alignment process erases the fine-grained signal of the mutation.

disease variant analysis, and protein engineering. This inherent loss of information at the alignment stage explains why such architectures are fundamentally ill-suited for these precision tasks.

# G    GENERALIZABILITY ACROSS BIOMOLECULAR TYPES

To demonstrate the broad applicability and robustness of our context-driven methodology, we evaluated its performance on standard benchmarks beyond our primary QA dataset. This tests the approach on different tasks and different biomolecular types.

## G.1    PERFORMANCE ON PROTEIN FUNCTION PREDICATION

We first evaluate our approach on the protein classification tasks from the Mol-Instruction benchmark (Fang et al., 2024). This benchmark contains curated datasets for assessing molecular understanding. We specifically tested on three functionally distinct protein subsets: Catalytic Activity, General Function, and Protein Function. The performance is shown in Figure 10.

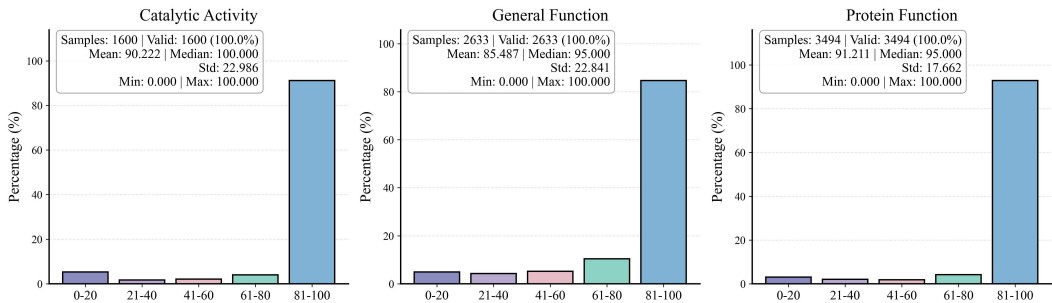

Figure 10:    Performance on Mol-Instruction Protein Classification. Score distributions of our context-driven method on three sub-datasets. The results show consistently high performance across all categories, with mean scores above 85 and the vast majority of answers ($> 84\%$) falling into the highest score bracket ($81 - 100$), demonstrating robust generalization.

As illustrated in Figure 10, our method achieves excellent performance across all three predication tasks. The mean score were exceptionally high: 91.2 for Protein Function, 90.2 for Catalytic Activity, and 85.5 for General Function. The score distributions are heavily skewed towards the maximum, with over 84% of answers in all three tasks receiving a score in the 81-100 range. This demonstrates that our method is not only effective on our QA benchmark but also generalizes robustly to standard, multi-category protein function predication tasks, validating its broad utility.

## G.2    PERFORMANCE ON DNA MUTATION PREDICTION

We evaluated two powerful generalist LLMs, GPT-4o and Qwen3-4B, across our three standard input configurations: Context-Only, Sequence + Context, and Sequence-Only. To provide a comprehensive view of performance, we measured not only classification accuracy but also F1 Score, Precision, and Recall, accounting for potential class imbalances in the dataset. The results, presented in Figure 11, strongly corroborate our findings from the protein-based tasks and confirm the paradigm's generalizability. For both models, the Context-Only configuration consistently achieved the highest scores across all four evaluation metrics. Crucially, the "informational noise" effect of raw sequences persists in the DNA domain. The Sequence + Context configuration consistently underperformed the Context-Only setup, indicating that the models were again "lost in tokenization," struggling to integrate the low-level signal from the raw DNA sequence with the clear, high-level context. The Sequence-Only approach yielded the poorest results, confirming that atomic tokenization of nucleotide sequences is insufficient for complex biological reasoning.

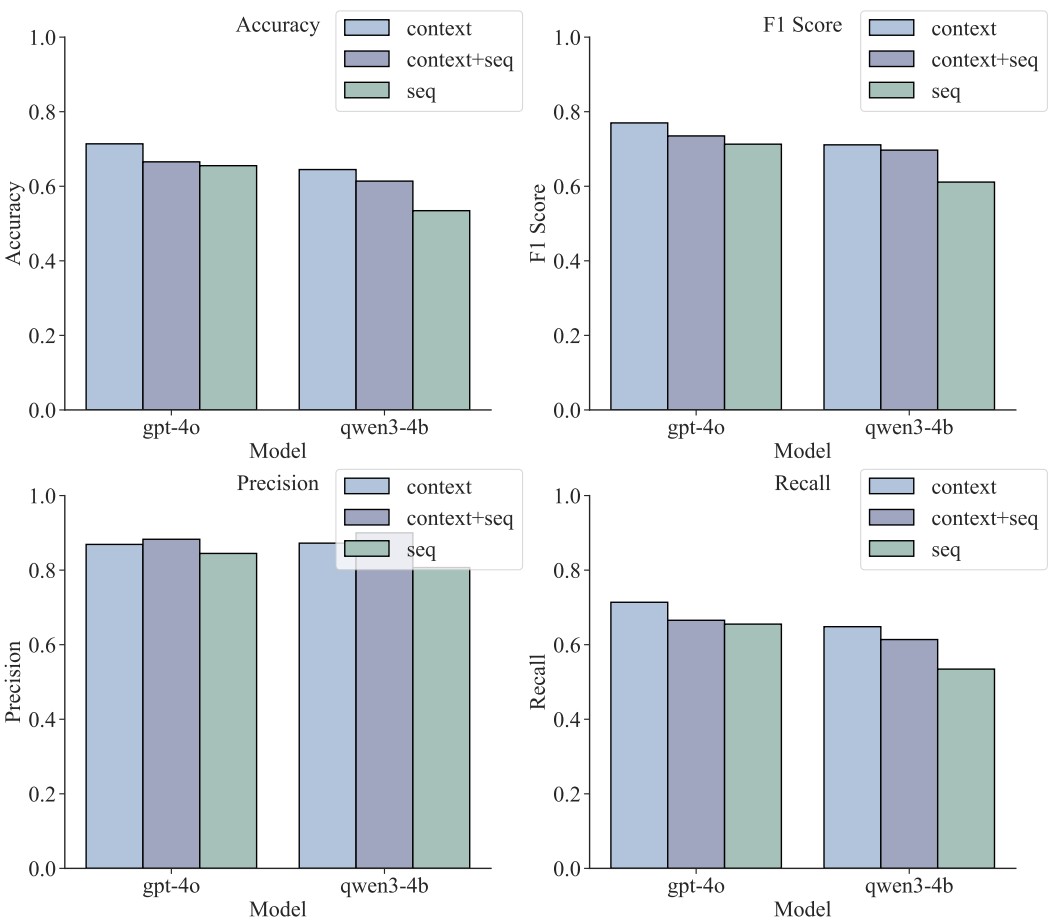

Figure 11: Model performance on the KEGG DNA dataset under different input configurations. Both Qwen3-4B and GPT-4o were evaluated across context-only, sequence-only, and combined inputs, with both models showing consistent improvements when using context-based inputs compared to sequence-based approaches.

## H    INDEPENDENCE FROM CLUSTERING METRICS

To verify that our conclusions are robust and not contingent on a single definition of protein families, we evaluated all embeddings against two distinct ground-truth labeling schemes: UniClust50 and UniClust30. These standards group proteins at 50% and a stricter 30% sequence identity threshold, respectively, providing different granularities for functional classification. While both are generated using the MMseqs2 algorithm, they represent conceptually different criteria for defining protein homology. We performed the same hierarchical clustering analysis for each representation against both sets of labels. The resulting ARI scores are presented in Table 4.

As shown in Table 4, the performance hierarchy and the specific ARI scores remain identical across both labeling schemes. While the identical results suggest that our test set's structure is stable across these two identity thresholds, the key insight is the unwavering consistency of our central observations. Specifically, three conclusions hold firm regardless of the ground-truth definition:

- Our context-driven approach consistently achieves the highest functional separation (ARI 0.958).

- The 'semantic misalignment' within Evolla is consistently demonstrated by the progressive drop in ARI from 0.945 to 0.809.

Table 4: Performance (ARI) against ground-truth labels from UniClust50 and UniClust30.

| Model / Representation Stage | ARI (vs. UniClust50) | ARI (vs. UniClust30) |
|---|---|---|
| Ours | **0.958** | **0.958** |
| Evolla Encoder (SaProt) | 0.945 | 0.945 |
| Evolla Alignment (Q-Former) | 0.916 | 0.916 |
| Evolla Decoder (Final) | 0.809 | 0.809 |
| Intern-S1 8B | 0.690 | 0.690 |
| NatureLM | 0.492 | 0.492 |

- The 'weak representation' of sequence-to-language models (Intern-S1 and NatureLM) is consistently evident from their significantly lower scores.

This stability across different conceptual standards for protein families validates our findings, confirming that the tokenization dilemma and the superiority of our context-driven solution are fundamental phenomena, not artifacts of a particular evaluation metric.

# I   QA EXAMPLES

To provide concrete illustrations of the performance differences reported in the main paper, this section presents qualitative examples from our benchmark. We analyze the outputs for a single protein (UniProt ID: `A6LHQ9`) across the three input modalities: Context-Only, Sequence-Only, and Sequence + Context. These case studies offer tangible insights into why our context-driven approach excels and how raw sequence inputs can be detrimental.

## I.1   COMPARATIVE ANALYSIS ON PROTEIN A6LHQ9

The following examples use the same protein and question ("What is the function of this protein?") to demonstrate how the input modality directly impacts the model's reasoning and final output. The ground truth is: "Putative component of the fimbrium tip. Fimbriae are filamentous appendages on the cell surface that mediate cell adhesion and biofilm formation".

### I.1.1   CASE 1: CONTEXT-ONLY (SUCCESS)

In the Context-Only setting, the model receives curated information from Pfam. The model demonstrates a strong ability to synthesize this structured knowledge into a comprehensive and accurate answer, achieving a perfect score.

---

**Protein A6LHQ9: Context-Only**

```
You are a senior systems biologist.  Analyze the input
information to answer the given question.
---------
Question:
What is the function of this protein?
---------
context Provided:
Conserved Domains (from Pfam):
• Pfam PF06321:  This family consists of several Porphyromonas
  gingivalis major fimbrial subunit protein (FimA) sequences.
  Fimbriae of Porphyromonas gingivalis, a periodontopathogen, play
  an important role in its adhesion to and invasion of host cells.
  The fimA genes encoding fimbrillin (FimA), a subunit protein
  of fimbriae, have been classified into five types, types I to
  V, based on nucleotide sequences.  It has been found that type
```

```
    II FimA can bind to epithelial cells most efficiently through
    specific host receptors [[cite:PUB00010404]].  Human dental
    plaque is a multispecies microbial biofilm that is associated
    with two common oral diseases, dental caries and periodontal
    disease.  There is an inter-species contact-dependent
    communication system between P. gingivalis and S. cristatus
    that involces the Arc-A enzyme [[cite:PUB00069820]].

• Pfam PF22449:  This domain is found at the C-terminal end of
    the putative fimbrium tip subunit Fim1F from Parabacteroides
    distasonis, which contain an N-terminal domain ([pfam:PF06321])
    and a slightly larger C-terminal domain (this entry) with a
    transthyretin-like fold that contains seven core beta-strands
    arranged in two beta-sheets an extra conserved 'appendage' of
    two amphipathic beta-strands [[cite:PUB00080711]].
    ---------
Answer:
The protein in question, which incorporates the domains
PF06321 (fimbrial subunit FimA) and PF22449 (C-terminal
transthyretin-like domain), functions as a structural component
of bacterial fimbriae with a primary role in adhesion to host
cells and surfaces.  Here's a detailed breakdown:

• PF06321 (FimA-like domain):  Found in fimbrial subunits of
  Porphyromonas gingivalis (a periodontal pathogen), this domain
  mediates host cell adhesion and invasion.  – Type II FimA (a
  variant within this family) binds efficiently to epithelial
  cells via specific host receptors, highlighting its role in
  pathogenicity and biofilm formation (e.g., in dental plaque).

• PF22449 (transthyretin-like domain):  Present at the C-terminus
  of fimbrial tip subunits (e.g., Fim1F in Parabacteroides
  distasonis), this domain stabilizes the fimbrial structure
  through its transthyretin-like fold (seven beta-strands forming
  two beta-sheets, plus an amphipathic ``appendage'').  – The
  transthyretin-like fold may enhance structural integrity
  or enable additional interactions, such as binding to host
  molecules or other bacterial proteins.

Overall Function:  The protein acts as a fimbrial structural
subunit, critical for assembling adhesive fimbriae.  These
appendages facilitate bacterial attachment to host tissues (e.g.,
oral epithelial cells) and surfaces, enabling colonization,
biofilm formation, and inter-species interactions (e.g., with
Streptococcus cristatus via the Arc-A system).  The combination
of PF06321 and PF22449 domains likely optimizes both adhesion
efficiency and structural stability, contributing to the
pathogenicity of periodontal bacteria.  Key References:  –
PF06321:  Binding to host receptors (PUB00010404).  – PF22449:
Structural role in fimbriae (PUB00080711).  – Inter-species
communication (PUB00069820).  ---------
Ground Truth:
Putative component of the fimbrium tip.  Fimbriae are filamentous
appendages on the cell surface that mediate cell adhesion and
biofilm formation.
    ---------
Result:
LLM-Score:  100
```

### I.1.2 CASE 2: SEQUENCE-ONLY (TOTAL FAILURE)

When provided with only the raw amino acid sequence, the model fails completely. It hallucinates a function (Chitin-Binding Module) that is entirely incorrect, likely by misinterpreting spurious patterns in the tokenized sequence. This is a stark example of the "lost in tokenization" phenomenon.

Protein A6LHQ9: Sequence-Only

You are a senior systems biologist. Analyze the input
information to answer the given question.
---------
**Question:**
What is the function of this protein?
---------
**Sequence Provided:**
MRFNVVLFMLIVALLGGLSTCSSEVPIGFDTDELSFDMSLVLLTGDMQTKASDPNYTYATTEEL
TIQNCHVAVFDKDGKRIYFKNFYSKDLGEMKTIGNLSGYELQLEGVRTFGKEDKKVSVLVVANA
NNANNSPFDNLTTYDGVDNSYTAKTIAKGPVTASLLVKIGKSETTLKYNQDNAPVTVSLIQLSA
KIEYTGVYKKENGELLEGFSLTKVAGLNASSKITIFNTSAVENGAFSDLAYPTTKPVTFYTYEI
SDAFKEVILSVQSGVEPKEYPFPANKFIKGNYYRIKGLKSSTEIEWVLENVEDKEVTLDPFE
---------
**Answer:**
Based on the amino acid sequence provided, this protein is **very
likely a Chitin-Binding Module (specifically, a CBM12 domain).**
Here's a breakdown of the evidence:

• **Signal Peptide:** The sequence starts with MRFNVLFLIVALLGGLSTCSS,
  which is a classic signal peptide. This indicates the protein
  is secreted from the cell.

• **CBM12 Signature Motifs:** The sequence contains the highly
  characteristic motifs of the Carbohydrate-Binding Module family
  12 (CBM12):

  − DGDG (appears twice: DGDG and DGDG).
  − NGAN (appears: NGAN).
  − TGS (appears: TGS).
  − QGN (appears: QGN).
  − Multiple conserved cysteine residues (C) that likely form
    stabilizing disulfide bonds.

• **Function of CBM12:**

  − **Primary Function:** Carbohydrate binding, specifically to
    N-acetylglucosamine (GlcNAc) residues. GlcNAc is the building
    block of chitin (a major component of fungal cell walls,
    insect exoskeletons, and crustacean shells) and is also found
    in various glycoproteins and glycolipids (like hyaluronan).
  − **Role in Proteins:** CBM12 domains are typically found
    attached to enzymes (like chitinases, chitobiosidases, or
    N-acetylglucosaminidases) or structural proteins. Their role
    is to target the catalytic domain to its specific substrate
    (chitin or other GlcNAc-containing polysaccharides) and
    increase the enzyme's efficiency by bringing it into close
    proximity with the substrate.
  − **Binding Specificity:** CBM12 modules generally bind to
    short stretches of GlcNAc residues, including chitobiose
    (GlcNAc-GlcNAc) and chitotriose (GlcNAc-GlcNAc-GlcNAc), and
    show weaker binding to isolated GlcNAc monomers. They do not
    bind cellulose.

In summary, the primary function of this protein is to act as
a Chitin-Binding Module (CBM12). Its job is to specifically
recognize and bind to N-acetylglucosamine (GlcNAc)-containing
carbohydrates, particularly chitin, facilitating the action
of associated enzymes (like chitinases) or contributing to the
structural integrity/function of proteins interacting with chitin
or related polysaccharides.
Typical Context: Proteins containing CBM12 domains are very
common in fungi, where they play crucial roles in cell wall

```
remodeling, nutrient acquisition (degrading chitin from other
organisms), and pathogenicity.  They are also found in bacteria
and some other organisms.
---------
Ground Truth:
Putative component of the fimbrium tip.  Fimbriae are filamentous
appendages on the cell surface that mediate cell adhesion and
biofilm formation.
---------
Result:
LLM-Score:  0
```

### I.1.3 CASE 3: SEQUENCE + CONTEXT (NOISE EFFECT)

In the combined setting, the model produces a correct but slightly less focused answer than the Context-Only version. The inclusion of the sequence acts as informational noise, causing a minor degradation in answer quality and a corresponding drop in the score. This supports our quantitative finding that the sequence is not merely unhelpful but can be actively detrimental.

---

**Protein A6LHQ9: Sequence + Context**

```
You are a senior systems biologist.  Analyze the input
information to answer the given question.
---------
Question:
What is the function of this protein?
---------
Context Provided:  [Same as Case 1]
Sequence Provided:  [Same as Case 2]
---------
Answer:
The protein in question, based on the sequence and associated
motifs (PF06321 and PF22449), functions as a fimbrial subunit
protein involved in bacterial adhesion and host-microbe
interactions.  Here's a detailed breakdown of its role:
• Fimbrial Structure and Assembly:  -The PF06321 domain
  (N-terminal) corresponds to the major fimbrial subunit FimA
  of Porphyromonas gingivalis.  This domain is critical for
  forming the structural backbone of fimbriae, which are hair-like
  appendages on the bacterial surface.  - The PF22449 domain
  (C-terminal) contains a transthyretin-like fold with a conserved
  ''appendage'' of amphipathic beta-strands.  This domain likely
  stabilizes the fimbrial tip structure and may mediate specific
  interactions with host cells or other microbial species.

• Adhesion and Invasion:  - Fimbriae in P. gingivalis (a
  periodontal pathogen) are essential for host cell adhesion
  and invasion.  The type II FimA variant (relevant to PF06321)
  binds efficiently to epithelial cells via specific host
  receptors, facilitating colonization and immune evasion.  -
  The transthyretin-like fold in PF22449 may enhance binding
  specificity or structural resilience, enabling the fimbriae
  to withstand mechanical stress in the oral environment.

• Biofilm Formation and Interspecies Communication:  - Fimbriae
  contribute to the formation of multispecies biofilms (e.g.,
  dental plaque), which are linked to oral diseases like
  periodontitis.  The protein may mediate interbacterial
  adhesion, as seen in the interaction between P. gingivalis
```

```
     and Streptococcus cristatus via the Arc-A enzyme system.  -
     The amphipathic beta-strand ``appendage'' in PF22449 could
     play a role in direct contact-dependent signaling or physical
     interactions with other microbes.  item Pathogenicity:  - By
     enabling adhesion to host tissues and other bacteria, this
     protein promotes the establishment of polymicrobial infections,
     contributing to the progression of periodontal disease and
     potentially systemic inflammation.
Conclusion:  The protein is a fimbrial subunit that assembles
into adhesive structures critical for bacterial attachment to
host cells, biofilm formation, and interspecies interactions
in the oral microbiome.  Its dual-domain architecture (PF06321
+ PF22449) ensures both structural integrity and functional
specificity in these processes.
---------
Ground Truth:
Putative component of the fimbrium tip.  Fimbriae are filamentous
appendages on the cell surface that mediate cell adhesion and
biofilm formation.

---------
Result:
LLM-Score:  95
```

The comparative analysis of protein `A6LHQ9` clearly illustrates the core tenets of our findings: context is the primary driver of accurate biological reasoning, while raw sequence data can lead to catastrophic failures or introduce performance-degrading noise.

## I.2 SHOWCASE OF OUR METHOD'S SUCCESS CASES

Having established the superiority of the context-driven paradigm, we now showcase its robustness and versatility across the three primary query types in our benchmark: molecular function, metabolic pathway, and subcellular localization. The following examples demonstrate the model's ability to consistently generate high-quality, accurate, and detailed answers for each category.

### I.2.1 EXAMPLE 1: FUNCTION PREDICTION

**The example of PDB ID: P0DJ91**

```
You are a senior systems biologist.  Analyze the input
information to answer the given question.
---------
Question:
What is the function of this protein?
---------
Conserved Domains (from Pfam):
• PF03549:  Intimin and its translocated intimin receptor
  (Tir) are bacterial proteins that mediate adhesion between
  mammalian cells and attaching and effacing (A/E) pathogens.
  A unique and essential feature of A/E bacterial pathogens is
  the formation of actin-rich pedestals beneath the intimately
  adherent bacteria and localised destruction of the intestinal
  brush border.  The bacterial outer membrane adhesin, intimin,
  is necessary for the production of the A/E lesion and diarrhoea.
  The A/E bacteria translocate their own receptor for intimin,
  Tir, into the membrane of mammalian cells using the type III
  secretion system.  The translocated Tir triggers additional host
  signalling events and actin nucleation, which are essential for
  lesion formation [[cite:PUB00006623]].  This family represents
  the Tir intimin-binding domain (Tir IBD) which is needed to
```

bind intimin and support the predicted topology for Tir, with both N- and C-terminal regions in the mammalian cell cytosol [[cite:PUB00014294]].

- **PF07489:** Intimin and its translocated intimin receptor (Tir) are bacterial proteins that mediate adhesion between mammalian cells and attaching and effacing (A/E) pathogens. A unique and essential feature of A/E bacterial pathogens is the formation of actin-rich pedestals beneath the intimately adherent bacteria and localised destruction of the intestinal brush border. The bacterial outer membrane adhesin, intimin, is necessary for the production of the A/E lesion and diarrhoea. The A/E bacteria translocate their own receptor for intimin, Tir, into the membrane of mammalian cells using the type III secretion system. The translocated Tir triggers additional host signalling events and actin nucleation, which are essential for lesion formation [[cite:PUB00006623]]. This family represents the Tir C-terminal domain which has been reported to bind uninfected host cells and beta-1 integrins although the role of intimin binding to integrins is unclear. This intimin C-terminal domain has also been shown to be sufficient for Tir recognition [[cite:PUB00014294]].

- **PF07490:** Intimin and its translocated intimin receptor (Tir) are bacterial proteins that mediate adhesion between mammalian cells and attaching and effacing (A/E) pathogens. A unique and essential feature of A/E bacterial pathogens is the formation of actin-rich pedestals beneath the intimately adherent bacteria and localised destruction of the intestinal brush border. The bacterial outer membrane adhesin, intimin, is necessary for the production of the A/E lesion and diarrhoea. The A/E bacteria translocate their own receptor for intimin, Tir, into the membrane of mammalian cells using the type III secretion system. The translocated Tir triggers additional host signalling events and actin nucleation, which are essential for lesion formation [[cite:PUB00006623]]. This family represents the Tir N-terminal domain which is involved in Tir stability and Tir secretion [[cite:PUB00014294]].

**Functional Annotations (from Homology via BLASTp):**
- GO terms associated with the homolog:
- GO term1: 0005515 · definition: Binding to a protein.

---------
**Answer:**
As a senior systems biologist, my analysis of the provided data leads to the following conclusion about the protein's function.
Summary of Function: Based on the provided information, this protein is the **Translocated Intimin Receptor (Tir)**, a key multi-domain effector protein from attaching and effacing (A/E) bacterial pathogens like enteropathogenic *E. coli* (EPEC). Its primary function is to act as a molecular bridge that subverts host cell processes to facilitate bacterial adhesion and pathogenesis. After being injected directly into the host cell by the bacterium, Tir inserts into the host cell membrane and functions as a high-affinity receptor for the bacterial outer membrane adhesin, **intimin**. This specific interaction is the central event that triggers a cascade of host signaling events, leading to dramatic cytoskeletal rearrangements and the formation of an ``actin pedestal'' structure beneath the bacterium.
Detailed Functional Analysis by Domain: The protein's overall function is accomplished through the coordinated action of its distinct domains:
1. **Tir Intimin-Binding Domain (IBD) (PF03549):** This is the core interaction domain. Its specific function is to **bind directly**

**to intimin**. This binding event is essential for the intimate attachment of the bacterium to the host cell and for establishing the correct topology of the Tir protein within the host membrane.
2. **Tir C-terminal Domain (PF07489):** This domain is also crucial for host interaction. It is sufficient for **recognizing and binding intimin**, reinforcing the function of the IBD. Additionally, it has a secondary role in **binding to host cell beta-1 integrins**, suggesting it may modulate or interfere with native host cell adhesion and signaling pathways, although the precise role of this interaction is noted as unclear.
3. **Tir N-terminal Domain (PF07490):** This domain's function is related to the protein's own lifecycle and deployment. It is involved in ensuring the **stability of the Tir protein** and is necessary for its **secretion** from the bacterium into the host cell via the Type III secretion system.
The associated Gene Ontology term, **GO:0005515 (protein binding)**, serves as a high-level functional annotation that is strongly supported by the specific binding activities described for both the IBD (PF03549) and the C-terminal domain (PF07489).
Role in the Biological System: From a systems perspective, Tir is a masterful example of pathogenic manipulation. The pathogen does not rely on a pre-existing host receptor; instead, it injects its own custom receptor into the host. The process is as follows:
1. The A/E pathogen uses its Type III Secretion System to inject Tir into a mammalian intestinal cell. 2. The Tir protein, guided by its N- and C-terminal domains, inserts into the host cell membrane, exposing its central intimin-binding domain to the exterior. 3. Intimin on the bacterial surface binds with high affinity to the translocated Tir. 4. This binding event triggers host signaling pathways, leading to the recruitment and nucleation of actin filaments directly beneath the bacterium.
5. This results in the formation of an ''actin-rich pedestal'' and the destruction of the local intestinal microvilli (the A/E lesion), which is the hallmark of this pathogenic mechanism and is critical for causing disease.
In summary, Tir is a multi-functional protein that acts as a central hub in bacterial pathogenesis. It ensures its own stability and secretion, inserts into the host membrane, and orchestrates the specific binding and signaling events required to hijack the host's cytoskeleton for the bacterium's benefit.

---------

**Ground Truth:**
Multifunctional protein that is required for efficient pedestal formation in host epithelial cells during infection. The extracellular region acts as a receptor for bacterial intimin, allowing the bacterium to attach tightly to the host-cell surface. Simultaneously, the intracellular region initiates a signaling cascade in the host cell, which leads to actin polymerization and formation of actin pedestals at the sites of bacterial adhesion (By similarity)

### I.2.2 EXAMPLE 2: PATHWAY PREDICTION

> **The example of PDB ID: P78748**
>
> You are a senior systems biologist. Analyze the input information to answer the given question.
> ---------
> **Question:**

```
What is the pathway of this protein?
---------
```
**Conserved Domains (from Pfam):**
```
• PF00215:  This family includes Orotidine 5'-phosphate
  decarboxylase enzymes [ec:4.1.1.23] that are involved in the
  final step of pyrimidine biosynthesis.  The family also includes
  enzymes such as hexulose-6-phosphate synthase.  This family
  appears to be distantly related to [pfam:PF00834].
```
**Functional Annotations (from Homology via BLASTp):**
```
- GO terms associated with the homolog:
```
```
• GO 0044205:  The chemical reactions and pathways resulting in
  the formation of UMP, uridine monophosphate, starting with
  the synthesis of (S)-dihydroorotate from bicarbonate; UMP
  biosynthesis may either occur via reduction by quinone, NAD+
  or oxygen.
```
```
• GO 0005829:  The part of the cytoplasm that does not contain
  organelles but which does contain other particulate matter, such
  as protein complexes.
```
```
• GO 0004590:  Catalysis of the reaction:  H+ + orotidine
  5'-phosphate = CO2 + UMP.
```
```
• GO 0006207:  The chemical reactions and pathways resulting in
  the formation of pyrimidine nucleobases, 1,3-diazine, organic
  nitrogenous bases, beginning with the synthesis of a pyrimidine
  ring from simpler precursors.
```
```
---------
```
**Answer:**
```
Based on a systems-level analysis of the provided data, the
protein is a key enzyme in the de novo pyrimidine biosynthesis
pathway.
Here is a breakdown of the evidence:
1.  Broad Pathway Identification:  The motif (PF00215) and
GO term GO:0006207 both explicitly place the protein within
''pyrimidine biosynthesis.''  This is the overarching metabolic
network responsible for creating pyrimidine nucleobases.
2.  Specific Sub-Pathway and Function:  The protein's role is
further refined to the UMP biosynthesis sub-pathway (GO:0044205).
The motif describes its function as the ''final step'' in this
process.
3.  Molecular Mechanism:  Its precise enzymatic function is
Orotidine 5'-phosphate decarboxylase activity (GO:0004590),
catalyzing the conversion of orotidine 5'-phosphate into UMP.
This confirms its role as the terminal enzyme in the synthesis of
UMP, a foundational pyrimidine nucleotide.
4.  Cellular Location:  The pathway is localized to the cytosol
(GO:0005829).
Conclusion:This protein is an Orotidine 5'-phosphate
decarboxylase that executes the final, decisive step of the UMP
biosynthesis pathway.  This pathway is a critical component of
the larger pyrimidine biosynthetic network, and it occurs within
the cytosol of the cell.
---------
```
**Ground Truth:**
```
Pyrimidine metabolism; UMP biosynthesis via de novo pathway; UMP
from orotate:  step 2/2
```

### I.2.3 EXAMPLE 3: SUBCELLULAR LOCATION PREDICTION

---

**The example of PDB ID: P63171**

```
You are a senior systems biologist.  Analyze the input
information to answer the given question.
---------
Question:
What is the subcellular location of this protein?
---------
Conserved Domains (from Pfam):
• PF03645:Tctex-1 is a dynein light chain.  It has been shown
  that Tctex-1 can bind to the cytoplasmic tail of rhodopsin.
  C-terminal rhodopsin mutations responsible for retinitis
  pigmentosa inhibit this interaction.
---------
Answer:
Based on the analysis of the provided information, the
subcellular location of Tctex-1 is the cytoplasm.
Here is the reasoning:

1. Component of the Dynein Motor Complex:  The protein is
   identified as a ``dynein light chain.''  Dynein is a motor
   protein complex that functions within the cytoplasm, moving
   cellular cargo along microtubule tracks.

2. Specific Binding Partner:  The text explicitly states that
   Tctex-1 binds to the cytoplasmic tail of rhodopsin.  For this
   interaction to occur, Tctex-1 must be located in the cytoplasm
   where it can access this specific domain of rhodopsin.

---------
Ground Truth:
Golgi apparatus, Cytoplasm, cytoskeleton, spindle
```

---

## J   LIMITATIONS

While our context-driven approach demonstrates impressive performance in various bioinformatics tasks, there are still significant challenges, particularly in predicting the effects of protein mutations. One of the primary difficulties is that InterProScan and BLAST—two key bioinformatics tools that we use for generating functional context—are not sensitive to subtle mutations at a single or few amino acid positions. This is because both tools rely on sequence-level comparisons, and mutations that alter a few amino acids may not result in any meaningful changes in the overall sequence context, making the corresponding Pfam domains or GO terms identical before and after mutation.

Thus, when we apply our method to predicting the effects of mutations, we encounter a situation where the context generated for the wild-type and mutated proteins is essentially the same. This leads to the limitation that our approach, at present, cannot effectively predict changes in the protein's function or characteristics due to small mutations.

Below are two examples demonstrating this limitation: one shows the context for a wild-type protein, and the other for a mutated version of the same protein. The only difference between the two sequences is the mutation at two amino acid positions, which we have highlighted in **red**. For clarity and ease of comparison, we have provided only the Pfam domain and GO annotations (numbers) rather than the complete context, which would otherwise be too long to display for these examples.

## J.1    WILD-TYPE PROTEIN EXAMPLE

---
**WT - Wild-Type**

**Conserved Domains (from Pfam):**
• **PF00732**

• **PF05199**
**Functional Annotations (from Homology via BLASTp):**
• **GO 0005737**

• **GO 0005576**

• **GO 0046562**

• **GO 0050660**

• **GO 0044550**

**Sequence:**
GIEASLLTDPKEVAGRTVDYIIAGGGLTGLTTAARLTENPDITVLVIESGSYES
DRGPIIEDLNAYGDIFGSSVDHAYETVELATNNQTALIRSGNGLGGSTLVNGGT
WTRPHKAQVDSWETVFGNEGWNWDSVAAYSLQAERARAPNAKQIAAGHYFNASC
HGINGTVHAGPRDTGDDYSPIVKALMSAVEDRGVPTKKDLGCGDPHGVSMFPNT
LHEDQVRSDAAREWLLPNYQRPNLQVLTGQYVGKVLLSQNATTPRAVGVEFGTH
KGNTHNVYAKHEVLLAAGSAVSPTILEYSGIGMKSILEPLGIDTVVDLPVGLNL
QDQTTSTVRSRITSAGAGQGQAAWFATFNETFGDYTEKAHELLNTKLEQWAEEA
VARGGFHNTTALLIQYENYRDWIVKDNVAYSELFLDTAGVASFDVWDLLPFTRG
YVHILDKDPYLRHFAYDPQYFLNELDLLGQAAATQLARNISNSGAMQTYFAGET
IPGDNLAYDADLRAWTEYIPYNFRPNYHGVGTCSMMPKEMGGVVDNAARVYGVQ
GLRVIDGSIPPTQMSSHVMTVFYAMALKIADAVLADYASMQ

---

## J.2    MUTATED PROTEIN EXAMPLE

---
**MUT1 - Mutated**

**Conserved Domains (from Pfam):**
[same as WT]
**Functional Annotations (from Homology via BLASTp):**
[same as WT]
**Sequence:**
GIEASLLTDPKEVAGRTVDYIIAGGGLTGLTTAARLTENPDITVLVIESGSYES
DRGPIIEDLNAYGDIFGSSVDHAYETVCLATNNQTALIRSGNGLGGSTLVNGGT
WTRPHKAQVDSWETVFGNEGWNWDSVAAYSLQAERARAPNAKQIAAGHYFNASC
HGINGTVHAGPRDTGDDYSPIVKALMSAVEDRGVPTKKDLGCGDPHGVSMFPNT
LHEDQVRSDAAREWLLPNYQRPNLQVLTGQYVGKVLLSQNATTPRAVGVEFGTH
KGNTHNVYAKHEVLLAAGSAVSPTILEYSGIGMKSILEPLGIDTVVDLPVGLNL
QDQTTSTVRSRITSAGAGQGQAAWFATFNETFGDYTEKAHELLNTKLEQWAEEA
VARGGFHNTTALLIQYENYRDWIVKDNVAYSELFLDTAGEASFDVWDLLPFTRG
YVHILDKDPYLRHFAYDPQYFLNELDLLGQAAATQLARNISNSGAMQTYFAGET
IPGDNLAYDADLRAWTEYIPYNFRPNYHGVGTCSMMPKEMGGVVDNAARVYGVQ
GLRVIDGSIPPTQMSSHVMTVFYAMALKIADAVLADYASMQ

---

In both cases, the context generated for the wild-type and mutated proteins is identical, as Inter-ProScan and BLAST fail to identify significant differences caused by mutations at amino acid positions. This limitation underscores the challenge of predicting the effects of mutations based on sequence context alone, as even small changes in the sequence can have profound effects on protein function that are not captured by these tools.

### J.3   SUMMARY OF THE LIMITATION

The core limitation of our current method is its inability to distinguish the functional consequences of amino acid mutations, due to the lack of sensitivity of InterProScan and BLAST to small sequence changes. This highlights the complexity of mutation prediction tasks, where even subtle changes to the sequence can lead to vastly different biological outcomes.

## K   COMPARISON OF MODEL PERFORMANCE ON THE SAME PROTEIN AND QUESTION

In this section, we compare the performance of different models on the task of predicting the function of the same protein, specifically protein **A7YWC4**, using the question: "What is the function of this protein" The ground truth for this protein is as follows:

---

**Ground Truth**

**Ground Truth:** Essential for mitochondrial network organization, mitochondrial metabolism and cell growth at the organism and cellular level. May play an important role in mitochondrial protein synthesis, mitochondrial DNA replication, and contribute to nucleoid stability. Also involved in mitochondrial-mediated antiviral immunity and the mitochondrial DNA damage response, promoting signaling between damaged genomes and the mitochondrial membrane, which leads to activation of the integrated stress response (ISR).

---

Below, we compare the answers generated by our method (using context), Interns1, Evolla, and NatureLM.

---

**Our Method (Context-Driven, Gemini2.5Pro)**

**Answer:**
As a senior systems biologist, my analysis of the provided information leads to the following conclusion about the function of this protein.
**Executive Summary**
This protein is **ATAD3 (ATPase family AAA domain-containing protein 3)**. It is a multi-functional mitochondrial protein that acts as a central hub for organizing mitochondrial structure and function. Its core activity is derived from its AAA ATPase domain, which uses the energy from ATP hydrolysis to perform chaperone-like functions, likely remodeling protein complexes. This molecular activity is essential for maintaining the mitochondrial network's architecture, regulating mitochondrial metabolism, and ultimately, supporting cellular growth.
**Detailed Functional Analysis**
The protein's function can be broken down by integrating the motif data with the relevant Gene Ontology (GO) terms.
**1. Protein Identity and Core Mechanism**
• **Identity:** The presence of the **PF12037** domain specifically identifies this protein as **ATAD3**.

• **Mechanism:** The protein contains a **PF00004** (AAA ATPase) domain. This domain confers two key molecular functions:

  – **ATP binding (GO:0005524):** As an ATPase, it must first bind ATP.
  – **ATPase activity (GO:0016887):** It hydrolyzes ATP to ADP to release energy. This energy powers its primary role as a **chaperone-like protein** that assists in the "assembly, operation, or disassembly of protein complexes."
**2. Biological Processes**
The energy-dependent chaperone activity of ATAD3 is leveraged to control several critical cellular processes:

• **Mitochondrion organization (GO:0007005):** This is a primary function. The PF12037 motif description explicitly states that ATAD3 is "essential for mitochondrial network organisation." This involves managing the shape, size, and distribution of mitochondria within the cell.

---

- **Regulation of cell growth (GO:0001558):** The motif information directly links ATAD3's function to being "essential for... cell growth at organism and cellular level." Healthy mitochondrial function is a prerequisite for cell proliferation.

- **Mitochondrial Metabolism and Protein Synthesis:** The PF12037 description notes its importance in "mitochondrial metabolis" and a potential role in "mitochondrial protein synthesis." This is consistent with its organizational role, as proper structure is required for efficient metabolic function and positioning of ribosomes. Its association with the **mitochondrial nucleoid (GO:0042645)**, where mitochondrial DNA resides, would be critical for these processes.

**3. Cellular Component (Localization)**
The protein's functions are tied to its specific location within the cell:

- **Mitochondrion (GO:0005739):** The protein is fundamentally mitochondrial.

- **Mitochondrial Membranes:** The PF12037 domain "interacts with the inner surface of the outer mitochondrial membrane." This positions the protein at a critical interface. Given its role in organizing the entire organelle, it is highly likely to also associate with the **mitochondrial inner membrane (GO:0005743)**.

- **Membrane Contact Sites (GO:0044233):** ATAD3's role in organizing the mitochondrial network and its localization at the membrane interface strongly suggest its presence at **endoplasmic reticulum-mitochondrion membrane contact sites**. These sites are crucial for inter-organelle communication and lipid exchange, which are vital for mitochondrial biogenesis and dynamics.

**Integrated Synopsis**
ATAD3 functions as a molecular machine anchored to the mitochondrial membranes. It uses the energy from ATP hydrolysis to remodel protein structures, thereby orchestrating the complex architecture of the mitochondrial network. By controlling the physical organization of the mitochondria, including its association with the nucleoid and contact sites with the ER, ATAD3 governs essential downstream processes. This structural regulation is indispensable for efficient mitochondrial metabolism, protein synthesis, and, consequently, for the overall growth and viability of the cell. Any disruption in ATAD3 function would lead to a collapse of the mitochondrial network, metabolic failure, and impaired cell growth.
**Score: 100**

---

### Interns1 (Sequence Input)

**Answer:**
Essential for mitochondrial network organization, mitochondrial metabolism and cell growth at organism and cellular level. May play an important role in mitochondrial protein synthesis. May also participate in mitochondrial DNA replication. May bind to mitochondrial DNA D-loops and contribute to nucleoid stability. Required for enhanced channeling of cholesterol for hormone-dependent steroidogenesis. Involved in mitochondrial-mediated antiviral innate immunity. Also involved in the mitochondrial DNA damage response by promoting signaling between damaged genomes and the mitochondrial membrane, leading to activation of the integrated stress response (ISR).

**LLM Answer:** The protein described by the provided amino acid sequence is likely an **ATPase** involved in energy-dependent cellular processes. Here's a detailed breakdown of its inferred function:

1. **ATPase Domain**:

   - The sequence contains a **Walker A motif** (GXXXXGKT), a hallmark of ATP-binding sites, and a potential **Walker B motif** (DExxG), which is critical for ATP hydrolysis. These motifs indicate the protein uses ATP hydrolysis to drive conformational changes or mechanical work.

2. **Structural Features**:

   - **Coiled-coil regions**: Suggest oligomerization or protein-protein interactions, common in ATPases that form multimeric complexes (e.g., hexamers).
   - **N-terminal proline/glycine-rich region**: May serve as a flexible linker or binding site for regulatory proteins or substrates.

3. **Functional Inference**:

- The protein is likely involved in **energy-dependent processes** such as:
  - **Protein remodeling/disaggregation** (e.g., chaperones like Hsp104).
  - **Vesicle trafficking** (e.g., NSF, which disassembles SNARE complexes).
  - **DNA repair/replication** (e.g., helicases that unwind DNA).
  - **Proteolysis** (e.g., subunits of ATP-dependent proteases like Lon or Clp).

4. **Key Motifs and Comparisons**:

- The Walker A/B motifs align with **AAA+ ATPases**, a family involved in diverse functions like protein degradation, membrane fusion, and DNA metabolism.
- The absence of protease-specific motifs (e.g., Ser-Lys dyad in Lon) or helicase signatures (e.g., DExD in DEAD-box proteins) makes these less likely, though not excluded.

**Conclusion**: The protein is most likely a **AAA+ ATPase** functioning in **protein complex remodeling** or **vesicle trafficking**, with roles in processes such as SNARE complex disassembly (like NSF) or stress response (like Hsp104). Further experimental validation (e.g., domain mapping, interaction studies) would refine its specific role.
**Score: 30**

---

### Evolla (Sequence Input)

**Answer:**
This protein, which is associated with the mitochondrial membrane in the organism Oryza sativa (rice), serves a multifaceted role primarily within the mitochondria. It is known as the Mitochondrial import receptor for the endoplasmic reticulum (MIA40), and its functions are intricately tied to the protein import and quality control processes within the mitochondria.

The primary function of this protein is to act as a receptor for the mitochondrial presequence translocase (MPT), which is crucial for the import of proteins into the mitochondrial matrix. This process is essential for the proper functioning of the mitochondria, as it ensures that the necessary proteins are correctly targeted and translocated into the organelle.

Additionally, the protein is involved in the translocation of proteins into the mitochondrial matrix. It is likely to recognize mitochondrial presequences that are less than 10 amino acids long, which are critical for the proper targeting and import of these proteins. This suggests a specialized role in the import of a subset of mitochondrial proteins.

The protein also plays a role in regulating the mitochondrial permeability transition pore (mPTP), which is a non-selective pore that can lead to mitochondrial swelling and cell death if activated. The regulation of this pore is vital for maintaining mitochondrial homeostasis and preventing apoptosis.

In terms of its interaction with other proteins, the protein forms homohexamers, and it is part of a complex that includes the MICOS complex subunit MIC10. This complex is involved in the regulation of the mitochondrial membrane potential and the opening of the mPTP. The protein's interaction with the mPTP is also noted to be indirect, suggesting a complex regulatory network within the mitochondria.

Moreover, the protein interacts with the mitochondrial presequence translocase, which is a hetero-hexamer composed of subunits TIM10, TIM22, and TIM23. This interaction is essential for the import of proteins into the mitochondrial matrix.

The protein's involvement in various cellular processes is further highlighted by its association with the biological process of protein import into the mitochondrial matrix, as well as its role in the regulation of mitochondrial membrane potential and the opening of the mPTP. These functions are integral to maintaining mitochondrial homeostasis and preventing cell death, making the protein a key player in cellular survival and metabolic regulation.
**Score: 30**

---

### NatureLM (Sequence Input)

**Answer:**
Upon analysis of the given protein sequence, it appears that the primary function of this protein is RNA binding, and it could be implicated in the mRNA transport.
**Score: 30**

### K.1 ANALYSIS OF MODEL DIFFERENCES

Our Method provided the most accurate and complete functional analysis, correctly identifying the protein as **ATAD3** and detailing its involvement in mitochondrial network organization, metabolism, and cell growth, which closely aligns with the ground truth. The score of 100 reflects the comprehensive nature of the answer and the high alignment with the expected protein function.

Interns1, relying solely on sequence input, inferred the protein to be an ATPase, which is a reasonable prediction given the presence of specific ATP-binding motifs. However, it failed to identify the specific protein (ATAD3) and did not connect its functions to the mitochondrial network organization, leading to a much lower score of 30.

Evolla also struggled with a correct protein identification, suggesting a mitochondrial import receptor for the endoplasmic reticulum (MIA40), which does not match the true function of ATAD3. This error resulted in a score of 20.

NatureLM provided a very generic answer, linking the protein to RNA binding and mRNA transport, which is not at all related to the actual function of ATAD3. This misinterpretation also earned a score of 20.

In conclusion, while all models gave some plausible biological functions, none of them fully captured the detailed and specific roles of ATAD3 within the mitochondrial network, as outlined in the ground truth. Our method, leveraging context, was able to provide the most accurate and thorough analysis of the protein's function, demonstrating the advantage of context-driven approaches over sequence-based models in protein functional prediction tasks.

## L THE USE OF LARGE LANGUAGE MODELS (LLMS)

In this work, we employed a Large Language Model (LLM) as an assistive tool to enhance the quality of our writing. Specifically, the LLM was utilized to identify and correct grammatical errors, as well as to refine the phrasing and style of the text to improve clarity and readability. Additionally, the LLM assisted in formatting the prompts within the "QA examples" section to ensure consistency and alignment with our intended experimental setup.

In accordance with the ICLR policies on LLM usage, we disclose our use of the LLM in this section and affirm that the assistance provided by the LLM did not compromise the originality or integrity of our work. We take full responsibility for the content of this manuscript and have ensured that all information is accurate and appropriately referenced.

## M DETAILED COST ESTIMATION

To ensure transparency, we provide the basis for our cost estimations using publicly available on-demand pricing and API costs.

**Settings**

- **CPU Instance (Single): An economical instance such as `c6a.xlarge`) at $0.153 / hour.**
- **CPU Instance (Batch): A powerful instance like `c6a.24xlarge` at $2.7 (each is 1.35) / hour.**
- **GPU Instance (Single A100): A single A100 instance at $2.75 / hour.**
- **LLM API Cost: A single call to DeepSeek-V3 is estimated at $0.000446.**

**Single-Sequence Inference**

- **Our Method Cost: ≈$0.0030**
  1. **CPU Cost: ($0.153 / 3600 seconds) * 60 seconds ≈ $0.00255.**
  2. **API Cost: $0.000446.**
  3. **Total: $0.00255 + $0.000446 = $0.002996 ≈ $0.0030.**

- **Evolla Cost:** ≈$0.0690
  1. **GPU Cost: ($2.75 / 3600 seconds) * 90 seconds ≈ $0.06875 ≈ $0.0690.**

**Batch Processing Inference**   The per-sequence averages for batch processing were derived by modeling a large-scale run to capture amortization and throughput effects.

- **Our Method: ≈$0.00054 per sequence**
  1. **Throughput Model: 2 powerful CPU machines process 1.12M sequences in 40 hours.**
  2. **CPU Cost: (2 machines * 40 hours * $1.35/hour) / 1.12M seq ≈ $0.000096/seq.**
  3. **API Cost: $0.000446/seq.**
  4. **Total: $0.000096 + $0.000446 = $0.00054/seq.**
- **Evolla: ≈$0.0152 per sequence**
  1. **Throughput Model: 1 A100 machine (with 20x concurrency) processes ≈ 180 sequences per hour.**
  2. **GPU Cost: (1 hour * $2.75/hour) / 180 seq ≈ $0.0152/seq.**

## N   WET-LAB VALIDATION SAMPLES

### N.1   PROMPT

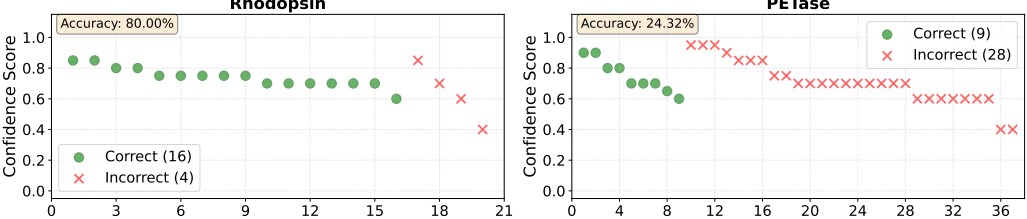

Figure 12:  Sample-level performance on classification accuracy of Intern-S1.

To evaluate the models' performance on the novel protein classification task, we designed a standardized prompt that frames the problem as a zero-shot binary classification. For each protein family under investigation (i.e., Rhodopsin or PETase), the model was tasked with determining whether the provided input protein belonged to the specified family or to a generic "Other" category. For the structure-aware model, Evolla, we first generated high-quality three-dimensional structure predictions for each novel sequence using the state-of-the-art AlphaFold3. It should be noted that the NatureLM model consistently failed to produce coherent or valid classifications across all samples in preliminary tests; it was therefore excluded from the final comparative analysis to maintain focus on viable models. Intern-S1's results are shown in Figure 12.

> **Prompt for Novel Protein Sequence Classification**
>
> What is the category of this protein?
> (1) Rhodopsin/PETase (2) Other
> Please provide your answer in the following structured format: Category: [1, 2] Confidence score: [a score between 0.0 and 1.0]

