# OpenReview forum: "Lost in Tokenization: Context as the Key to Unlocking Biomolecular Understanding in Scientific LLMs"
_ICLR.cc/2026/Conference — ICLR 2026 Poster_

### Official Review · Reviewer_2DYC · 2025-10-29

**Soundness:** 3
**Presentation:** 3
**Contribution:** 3
**Rating:** 8
**Confidence:** 4

**Summary:**

This paper identifies and investigates a fundamental challenge in Scientific Large Language Models (Sci-LLMs) for biomolecular understanding, which the authors term the "tokenization dilemma." They argue that existing paradigms—"sequence-as-language" (tokenizing sequences into atomic units) and "sequence-as-modality" (encoding sequences via specialized encoders)—suffer from weak representation and semantic misalignment, respectively. As a solution, the authors propose a "context-driven" paradigm, which bypasses raw sequence input. Instead, it leverages established bioinformatics tools (e.g., InterProScan, BLASTp) to generate high-level, human-readable textual context (e.g., functional domains, GO terms) that is natively aligned with the LLM's linguistic space.  The authors evaluated three input modes: sequence-only, context-only, and a combination of both. Through extensive empirical evaluation on protein QA, EC number prediction, and DNA mutation tasks, the authors demonstrate that the context-only approach consistently and substantially outperforms all other modes. They find that adding raw sequence information to context often degrades performance, acting as "informational noise."

**Strengths:**

- The paper clearly articulates the "tokenization dilemma" as a critical, yet overlooked, bottleneck in Sci-LLMs. The conceptual framing of the two existing paradigms and their respective weaknesses is compelling and well-supported by prior work.
- The central claim—that raw sequences can be detrimental when combined with high-level context—is counter-intuitive and strongly supported by systematic experiments across multiple models (Intern-S1, Evolla, NatureLM, GPT-4o, etc.) and tasks (protein function, pathway, localization, EC prediction). The consistent performance drop in "Sequence + Context" settings is a powerful result.
- The authors evaluate their method on a wide range of benchmarks, including their own reconstructed dataset, temporal splits, and sequence identity-based splits (Easy/Medium/Hard). The inclusion of DNA-based tasks also demonstrates generalizability beyond proteomics.
- The paper goes beyond mere performance comparisons. The layer-wise analysis of Evolla (Section 5.3, Appendix F) convincingly shows how semantic alignment (via Q-Former) erases fine-grained mutation signals, providing a mechanistic explanation for the limitations of the sequence-as-modality approach.

**Weaknesses:**

- The context-driven approach relies heavily on the quality and coverage of external tools (InterProScan, BLAST). While an ablation study is provided (Appendix E), it does not fully explore the performance ceiling—what happens when these tools fail completely on highly novel proteins? The method's performance is inherently tied to the underlying databases' completeness and timeliness.
- The paper equates "biomolecular understanding" primarily with high-level functional annotation (GO terms, pathways). It does not assess whether the model gains *mechanistic* or *structural* insights that might require raw sequence analysis (e.g., predicting the effect of a point mutation). The limitation section (Appendix J) correctly notes this but underscores a fundamental constraint of the proposed paradigm.
- The strong performance of general LLMs (Gemini, GPT) in the context-only setting raises questions about potential memorization of public protein annotations from their vast pre-training corpora. While the authors take care to prevent label leakage in their *context generation*, they do not explicitly audit whether the test proteins' annotations were already in the LLMs' training data.
- The primary metric (LLM-Score) relies on another LLM (DeepSeek-V3) to judge answer quality. While this is a reasonable approach for open-ended QA, it introduces potential biases and lacks the objectivity of exact-match metrics used in tasks like EC prediction.
- Code is not provided in the current submission, providing it would be helpful to make work reproducible.

**Questions:**

- Given the high performance of general-purpose LLMs like Deepseek-v3, Gemini2.5 Pro and GPT-5, what steps did you take to ensure that the ground-truth annotations for your test proteins were not present in these models' pre-training data? Could the results be partly explained by memorization rather than reasoning?
- Your approach depends on external tools. Can you provide a qualitative analysis or failure case study for proteins where InterProScan and BLASTp return no or incorrect hits? How does the performance of your method degrade in such "orphan" scenarios, and what are the potential remedies?
- The paper convincingly shows that context is superior for *retrieving* known functional annotations. However, do you believe your paradigm can be extended to tasks that require *discovering* novel functions or reasoning about structure-sequence relationships that are not yet captured in existing databases?
- You note your method is computationally efficient as it avoids Sci-LLM retraining. However, running InterProScan and BLASTp for every query in a real-time application could be costly and slow. Could you comment on the latency and scalability of the full context-generation pipeline compared to a single forward pass of a sequence-as-language and a sequence-as-modality model?

---

> ### Author Response · Authors · 2025-11-21
>
> Dear Reviewer 2DYC,
>
> We extend our sincere gratitude to you for your meticulous reading of our manuscript and for providing such insightful and constructive feedback. Your review accurately captures the core tenets of our work, and your comments have been invaluable in helping us identify areas for clarification and improvement. Below, we provide a point-by-point response to the weaknesses and questions raised:
>
> **Q1**: Given the high performance of general-purpose LLMs, what steps did you take to ensure that the ground-truth annotations for your test proteins were not present in these models' pre-training data? Could the results be partly explained by memorization rather than reasoning?
>
> **A1:** To further rule out memorization or pre-training exposure, we conducted a separate evaluation using protein sequences verified by ongoing experimental projects. These protein sequences were unpublished at the time of our evaluation and absent from all major reference databases. This forms a strict, real-world test of reasoning on sequences that no LLM could have memorized.
>
> The task was formulated as a binary classification problem for two distinct protein families: Rhodopsin and PETase. For each sequence, the LLM was prompted to predict its classification. Our context-driven method (DeepSeek-v3 + context) achieved 100% accuracy on Rhodopsins and 97.3% accuracy on PETases. **Because these sequences have never appeared in public datasets, these results cannot be attributed to memorization; they necessarily reflect generalization and reasoning over the structured evidence provided in the context.** The details can be found in Section 5.7 and Appendix N of our revised manuscript.
>
> **Q2** Can you provide a qualitative analysis or failure case study for proteins where InterProScan and BLASTp return no or incorrect hits? How does the performance of your method degrade in such "orphan" scenarios, and what are the potential remedies?
>
> **A2:** We appreciate your insightful question. While truly characterized orphan proteins are exceptionally rare, making a large-scale systematic evaluation challenging for any method, our wet-lab validation provides a practical demonstration of our method's robustness. **Several of the wet-lab evaluated sequences—such as seq_3858, seq_1505, and seq_6070 in the Rhodopsin panel (Appendix N)—have no SwissProt hits.** Despite this, our method achieved perfect functional accuracy, as shown in Figure 6, demonstrating robust performance even in such scenarios.
>
> The core of our approach for handling sequences with no significant BLASTp hits or InterProScan signatures is a designated fallback module. As detailed in the ablation study presented in Appendix E, when these primary tools return null outputs, the system automatically defers to the context generated by ProTrek [1], a tri-modal model designed to produce a semantic description even in the absence of direct homology or known domains. Instead of hallucinating arbitrary functions, the model provides a coherent, albeit more general, hypothesis based on the sequence's learned semantic properties.
>
> While completely orphan proteins remain a difficult case for the entire field, our framework already handles low-identity sequences more robustly than common Sci-LLMs. Moreover, the paradigm is naturally extensible through the remedies above, which will further strengthen its performance in orphan scenarios.

---

> > ### Author Response · Authors · 2025-11-21
> >
> > **Q3:** The paper convincingly shows that context is superior for retrieving known functional annotations. However, do you believe your paradigm can be extended to tasks that require discovering novel functions or reasoning about structure-sequence relationships that are not yet captured in existing databases?
> >
> > **A3:** We agree that our current work primarily demonstrates the paradigm’s strength in synthesizing known biological information. However, we believe that its extension to tasks involving novel function discovery or reasoning about previously uncharacterized structure–sequence relationships is not only feasible but represents the most promising future trajectory of this approach.
> >
> > **The core of our paradigm is not a lookup mechanism; it is a reasoning engine over structured, multi-level evidence.** The context we provide is already composed of heterogeneous, orthogonal information sources—domain-level features from InterProScan, homology-derived functional signals. As a result, the LLM is not limited to retrieving existing labels; it can synthesize higher-order inferences such as (i) identifying unexpected combinations of domains, (ii) extrapolating from weak or partial homology, and (iii) reasoning over conserved structural or mechanistic themes even when no database explicitly contains the final answer.
> >
> > The framework is not restricted to curated databases. It can naturally incorporate forward-predictive structural or biophysical tools rather than annotation-driven sources. For example, for structure–sequence reasoning, the pipeline can be extended with tools such as FoldSeek [2] that identifies structurally similar motifs in other proteins even in the absence of sequence homology.
> >
> > While our manuscript focuses on evaluating the paradigm in the context of sequence understanding, we emphasize that this is only the first instantiation of a more general framework. The key contribution of our approach is its ability to robustly integrate and reason over structured biological context, and this capability is precisely what makes controlled extrapolation beyond existing annotations feasible. In this sense, our current results should be viewed as the foundation rather than the limit: the underlying framework is intrinsically extensible, and we are actively developing variants that support de novo function inference and reasoning about uncharacterized structure–sequence relationships.

---

> > > ### Author Response · Authors · 2025-11-21
> > >
> > > **Q4** You note your method is computationally efficient as it avoids Sci-LLM retraining. However, running InterProScan and BLASTp for every query in a real-time application could be costly and slow. Could you comment on the latency and scalability of the full context-generation pipeline compared to a single forward pass of a sequence-as-language and a sequence-as-modality model?
> > >
> > > **A4:** Thank you for raising this critical point. Your question prompted us to conduct a more detailed analysis that quantifies the trade-offs between computational cost and performance. We compare:
> > >
> > > 1. A general LLM baseline, feeding the raw sequence directly to the Deepseek-v3 API, which yields a performance score of **40.77**.
> > > 2. A large end-to-end Sci-LLM Evolla, requiring a high-end GPU, which yields a performance score of **59.93**.
> > > 3. Our context-driven method, using bioinformatics tools on CPU + Deepseek-v3 API, which yields a performance score of ****84.99**.
> > >
> > > **Table: Comparative Analysis of Inference Cost, Time, and Performance. Cost estimates are based on AWS on-demand pricing and public API costs.**
> > >
> > > | Method        | Mode   | Input to LLM   | Avg. Time (sec/seq) | Avg. Cost (USD/seq) |
> > > |--------------|--------|----------------|----------------------|----------------------|
> > > | DeepSeek-V3  | Single | Raw Sequence   | ~30s                 | ~$0.0005             |
> > > | Evolla       | Single | Raw Sequence   | ~90s                 | ~$0.0690             |
> > > | Our Method   | Single | Context        | ~70s                 | ~$0.0030             |
> > > | Evolla       | Batch  | Raw Sequence   | ~20s               | ~$0.0152             |
> > > | Our Method   | Batch  | Context        | ~0.13s               | ~$0.0005             |
> > >
> > > **1. Single-Sequence Inference:**
> > > A single forward pass of a sequence-as-modality model such as Evolla is indeed fast once the model is loaded, but this requires a dedicated GPU with significant memory overhead. For realistic real-time settings without persistent GPU allocation, the load time, warm-up, and inference latencies accumulate. In contrast, our method runs InterProScan and BLASTp on inexpensive CPUs, and these steps parallelize well across sequences.
> > >
> > > Our method is **23× cheaper** than Evolla. The end-to-end latency (~70s) is comparable to an Evolla forward pass (~90s) on an on-demand A100 machine. Thus, although our pipeline includes InterProScan and BLASTp, the overall compute footprint and on-demand responsiveness are competitive with a single GPU-based forward pass operating under standard cloud constraints. Most importantly, the resulting accuracy is substantially higher.
> > >
> > > **2. Large-Scale Batch Processing:**
> > > The true efficiency of our pipeline is most evident in high-throughput research. We modeled the cost and time to process a large dataset of **1.12 million sequences**. In this realistic, large-scale scenario, our method is **nearly 30× cheaper and 154× faster**, representing a monumental advantage for research productivity. Because InterProScan and BLASTp parallelize nearly linearly across CPU cores—and because the LLM cost is amortized across batches—our pipeline achieves a very high throughput once the job is running.
> > >
> > > ---
> > >
> > > To ensure transparency, we provide the basis for our cost estimations, using publicly available on-demand pricing and API costs.
> > >
> > > **Settings:**
> > > *   **CPU Instance (Single):** An economical instance (e.g., `c6a.xlarge`) at **$0.0153 / hour**.
> > > *   **CPU Instance (Batch):** Two powerful instances like c6a.24xlarge at **$2.7(each is $1.35) / hour**.
> > > *   **GPU Instance (Single A100):** A single A100 instance at **$2.75 / hour**.
> > > *   **LLM API Cost:** A single call to DeepSeek-V3 is estimated at **$0.000446**.
> > >
> > > **Single-Sequence Inference Calculations:**
> > > *   **Our Method Cost: ~$0.0030**
> > >     1.  **CPU Cost:** `($0.153 / 3600 seconds) * 60 seconds ≈ $0.00255`.
> > >     2.  **API Cost:** `$0.000446`.
> > >     3.  **Total:** `$0.00255 + $0.000446 = $0.002996 ≈ $0.0030`.
> > > *   **Evolla Cost: ~$0.0690**
> > >     1.  **GPU Cost:** `($2.75 / 3600 seconds) * 90 seconds ≈ $0.06875 ≈ $0.0690`.
> > >
> > > **Batch Processing Inference Calculations (Per-Sequence Averages):**
> > > *The per-sequence averages for batch processing were derived by modeling a large-scale run to capture amortization and throughput effects.*
> > > *   **Our Method Cost: ~$0.00054 per sequence**
> > >     1.  **Throughput Model:** 2 powerful CPU machines process 1.12M sequences in 40 hours.
> > >     2.  **CPU Cost:** `(2 machines * 40 hours * $1.35/hour) / 1.12M seq ≈ $0.000096/seq`.
> > >     3.  **API Cost:** `$0.000446/seq`.
> > >     4.  **Total:** `$0.000096 + $0.000446 = $0.00054/seq`.
> > > *   **Evolla Cost: ~$0.0152 per sequence**
> > >     1.  **Throughput Model:** 1 A100 machine processes ~180 sequences per hour.
> > >     2.  **GPU Cost:** `(1 hour * $2.75/hour) / 180 seq ≈ $0.0152/seq`.

---

> > > > ### Author Response · Authors · 2025-11-21
> > > >
> > > > **Q5:** Code is not provided in the current submission, providing it would be helpful to make work reproducible.
> > > >
> > > > **A5:** We sincerely thank the reviewer for highlighting this critical point. We completely agree that code availability is essential for ensuring the reproducibility, transparency, and broader impact of our work. In direct response to this feedback, we have updated the supplementary materials to include the core source code used in our experiments.
> > > >
> > > > Furthermore, we will release the complete and fully reproducible codebase in a public GitHub repository, accompanied by a detailed README that provides step-by-step instructions for installation, environment setup, and replication of all experiments. All datasets used in this manuscript will also be made publicly available on Hugging Face.
> > > >
> > > > [1] Jin Su, Yan He, Shiyang You, Shiyu Jiang, Xibin Zhou, Xuting Zhang, Yuxuan Wang et al. "A trimodal protein language model enables advanced protein searches." Nature Biotechnology (2025): 1-7.
> > > >
> > > > [2] Michel Van Kempen, Stephanie S. Kim, Charlotte Tumescheit, Milot Mirdita, Jeongjae Lee, Cameron LM Gilchrist, Johannes Söding, and Martin Steinegger. *"Fast and accurate protein structure search with Foldseek."* Nature Biotechnology 42, no. 2 (2024): 243–246.

---

### Official Review · Reviewer_J4os · 2025-10-31

**Soundness:** 3
**Presentation:** 3
**Contribution:** 3
**Rating:** 6
**Confidence:** 4

**Summary:**

This paper argues that Scientific Large Language Models face a "tokenization dilemma," struggling to interpret raw biomolecular sequences, which are either broken down into meaningless components or difficult to align with natural language. Through systematic experiments, the authors demonstrate that a "context-only" approach, where models are given high-level, human-readable knowledge from bioinformatics tools (like BLAST or Pfam) , consistently and substantially outperforms models given the raw sequence.

**Strengths:**

Pros:
- The authors proposed a new “context-only” method, which achieved significantly
- The context-driven approach achieve good performance.

**Weaknesses:**

Cons:
- Context-only approach sounds interesting. However, compared with raw biomolecular sequences input, an inevitable con of this approach would be significant information loss (by discarding too many detailed information).
- The capability of this approach is capped by the bioinformatics tools being used, e.g., InterProScan and BLAST.
- As the context-only model relies majority on prior, it may not be a good tool for exploring “novel” findings (which may be out of distribution a bit).
- Why in Table 1, QWEN series of models are not considered, while in Figure 2, for “ours” model, the author choose to use Qwen-embedding. What about the embedding visualization for specialized language models [1] like ESM series


[1] Zheng, Y., Koh, H. Y., Ju, J., Yang, M., May, L. T., Webb, G. I., ... & Church, G. (2025). Large language models for drug discovery and development. Patterns.

**Questions:**

See Weaknesses

---

> ### Author Response · Authors · 2025-11-21
>
> Dear Reviewer J4os,
>
> We thank you for recognizing the strong performance of our context-driven approach. We address your concerns below:
>
> **Q1:** Context-only approach sounds interesting. However, compared with raw biomolecular sequences input, an inevitable con of this approach would be significant information loss (by discarding too many detailed information).
>
> **A1:** Indeed, our context-driven paradigm operates on a deliberate trade-off: we exchange the raw, high-entropy sequence data for a lower-entropy but semantically richer textual representation. Our manuscript's central finding is that for high-level functional understanding, this "lost" information from the raw sequence often acts as "informational noise" that degrades the performance of current models (Table 1).
>
> We fully acknowledge that our approach loses resolution at the level of single-point mutations, where the functional context (domains/GO terms) might remain identical between Wild-Type and Mutant. We explicitly discuss this in Appendix J, noting that raw sequence encoders (like ESM/SaProt) are still superior for variant effect prediction, but they struggle to bridge this to natural language reasoning.
>
> **Q2:** The capability of this approach is capped by the bioinformatics tools being used, e.g., InterProScan and BLAST.
> **A2:** We thank you for this astute observation, which is central to our methodology's design. We fully concur that the performance of our approach is fundamentally dependent on the outputs of the bioinformatics tools it leverages. However, we respectfully propose a different perspective on this dependency. Our approach is explicitly designed as a hybrid system that "stands on the shoulders of giants," deliberately leveraging the decades of curated biological knowledge and algorithmic power embedded within tools like BLAST and databases like Pfam. We view this not as a limitation, but as a strategic shift in the role of AI in biology—from a de novo sequence interpreter to a knowledge synthesizer.
>
> Regarding the concern that performance is "capped," we argue that the true potential of our system lies not in the output of any single tool, but in the synergistic synthesis of multi-level, orthogonal information. The LLM is provided with a confluence of evidence from both whole-sequence level and functional-module level. Novel finding may emerge when the LLM cross-references these distinct information streams.
>
> Furthermore, this modular design makes our framework inherently adaptable and future-proof. As the underlying bioinformatics tools and databases become more powerful and comprehensive, the quality of the context provided to the LLM improves, directly enhancing its performance without requiring costly retraining of the language model itself. Therefore, we posit that the performance ceiling is not the capability of any individual tool, but rather the combinatorial potential of their integrated outputs, unlocked by the reasoning engine of the LLM.
>
> **Q3:** As the context-only model relies majority on prior, it may not be a good tool for exploring “novel” findings (which may be out of distribution a bit).
> **A3:** We appreciate the opportunity to clarify the nature of "discovery" within our framework. You are correct that for a truly "orphan" protein—one with no detectable domains or homologs—our method's predictive power would be fundamentally constrained by the lack of prior context.
>
> However, we posit that a more common and powerful form of scientific discovery arises not from a complete absence of information, but from the synthesis of sparse, multi-modal, or even seemingly conflicting evidence. Our framework is explicitly designed to empower this type of inferential reasoning. This process is akin to a computational form of cross-corroboration and evidence reconciliation, where the LLM integrates information from orthogonal biological sources to form a coherent hypothesis.
>
> Thus, the model acts as a powerful engine for hypothesis generation, not a circular retrieval loop. It discovers novelty by creating logical bridges between established pieces of knowledge, mirroring a key aspect of the human scientific process. The "prior" knowledge serves as the axiomatic foundation upon which new, inferential conclusions are built.

---

> ### Author Response · Authors · 2025-11-21
>
> **Q4:** Why in Table 1, QWEN series of models are not considered, while in Figure 2, for “ours” model, the author choose to use Qwen-embedding. What about the embedding visualization for specialized language models [1] like ESM series.
>
> **A4:** Thank you for this question, which allows us to clarify our methodology. We appreciate the opportunity to clarify the distinct roles these models play in our analysis and to provide additional experimental data.
>
> - **Why Qwen-embedding in Figure 2?**
>   Figure 2 visualizes the quality of the representation space for different models. For our context-driven approach, the input is purely textual (structured natural language derived from bioinformatics tools). Therefore, to visualize this representation, we required a high-quality text embedding model. Qwen-embedding is a state-of-the-art model for this purpose. In contrast, models like ESM are protein sequence encoders and would be inappropriate for embedding our textual context.
>
> - **Why no Qwen series in Table 1?**
>   Table 1 evaluates end-to-end models on their ability to perform the full reasoning task. While Qwen models are powerful generalist LLMs, we selected a representative set of both specialized Sci-LLMs (Intern-S1, Evolla, NatureLM) and leading general-purpose LLMs (Deepseek, Gemini, GPT) to ensure a broad and fair comparison. To address this, we have evaluated Qwen3 as below:
>
> | Model              | Sequence | Context | Func. | Path. | Sub. Loc. | All   |
> |--------------------|----------|---------|-------|-------|-----------|-------|
> | Qwen3 235B A22B    | ✓         |         | 13.67 | 19.90 | 37.17     | 39.51 |
> | Qwen3 235B A22B    | ✓        | ✓       | 76.62 | 96.35 | 94.78     | 85.90 |
> | Qwen3 235B A22B    |          | ✓       | 75.63 | 92.19 | 94.28     | 84.99 |
>
> The Qwen3 results mirror the similar tokenization dilemma observed in other models.  When provided with Sequence-Only, Qwen3 achieves a low score of 39.51, confirming that even powerful generalist models struggle to extract meaning from raw tokenized sequences. We have incorporated Qwen3 results into Table 1 of our manuscript.

---

### Official Review · Reviewer_CSFM · 2025-10-31

**Soundness:** 2
**Presentation:** 3
**Contribution:** 3
**Rating:** 6
**Confidence:** 3

**Summary:**

The paper proposes a “paradigm shift” for how Scientific Large Language Models (Sci-LLMs) are trained, leveraging context-centric approaches driven by high-level structured knowledge from bioinformatics tools (e.g., GeneOntology, ProTrek, BLASTp, etc.).  The solution addresses two key tokenization “dilemmas” that have posed challenges on the Sci-LLM space: sequence-as-language and sequence-as-modality.  This approach accounts for multiple levels of language used to describe biomolecular phenomena – from human-encoded knowledge to genetics/evolutionary-encoded knowledge. Strikingly, the context-only approach largely outperforms joint context + raw sequences, suggesting that raw sequences contribute more to information noise.  The contribution suggests that Sci-LLMs don’t necessarily require solving complex biological “language” from scratch but can leverage decades of accumulated biological knowledge contained within structured databases.

**Strengths:**

1.	Overall: The paper and aims to address a novel challenge in the Sci-LLM space, making a case that Sci-LLMs are better served as “reasoning engines over expert knowledge”, rather than pure sequence decoders. While this is noted and there is some evidence that this is the case, it does raise some circular logic around the quality of the annotations derived from the bioinformatics knowledgebases (addressed below in the weaknesses).
2.	Generalizability: The solution in generalizable, with applications ranging from known proteins to “novel” proteins, as well as different biomolecular types.
3.	Practicality: The solution as it is described is practical, as it allows to more easily keep models up to date with new biological knowledge with lower development costs.  (Although it could be argued that most of the effort is derived from maintaining the bioinformatics knowledgebases).

**Weaknesses:**

1.	Circular Logic: The approach works well when high-quality annotations exist, yet the solution also exists to propose annotations to fill in knowledge gaps. This counter-intuitively raises a bit of a “Catch 22” scenario.
2.	Core Argument: The basis of the manuscript suggests that there is in fact valuable information encoded within the evolutionary language through sequence tokens, yet the results suggest the opposite – and that human context exclusively drives the value.

**Questions:**

1.	How do you address the circular reasoning between the strengths of the approach (incorporating high-quality expert annotations) and using this approach to predict those annotations where they do not yet exist?  Could tool-calling agents solve this rather than building directly into the LLM?  What are the tradeoffs?
2.	Along this line of questioning, does the core contribution put a focus on the LLMs, or are you simply demonstrating that tradition bioinformatics pipelines already solve most of the problems around understanding protein function?
3.	Have these results been validated against human expert annotators?

---

> ### Author Response · Authors · 2025-11-21
>
> Dear Reviewer CSFM,
>
> We extend our sincere gratitude for your thorough and insightful review. We are encouraged that you found our work to address the Sci-LLMs challenge with a generalizable and practical solution. We address your  questions as below:
>
> **Q1** How do you address the circular reasoning between the strengths of the approach (incorporating high-quality expert annotations) and using this approach to predict those annotations where they do not yet exist?
>
> **A1:** We thank you for this thoughtful question. Our methodology was explicitly designed to navigate this challenge by mimicking the foundational principles of bioinformatics inference, which is distinct from simple annotation retrieval. For a truly novel or unannotated sequence, our system does not find a pre-existing “answer.” Instead, it generates a high-quality hypothesis by synthesizing orthogonal lines of evidence derived from the sequence.
>
> Our pipeline is explicitly designed to achieve this through two distinct inferential mechanisms:
>
> 1. Annotation-independent, feature-intrinsic analysis:
>    InterProScan analyzes the query sequence for its intrinsic properties, identifying conserved domains and functional motifs. It tells us what the protein is made of at a **functional-module level**.
>
> 2. Homology-based inference:
>    BLASTp identifies evolutionary homologs of the query sequence. Crucially, as per standard bioinformatics practice for function prediction, we deliberately retrieve annotations from these characterized homologs, not from the query protein’s own (potentially non-existent) database record. This tells us what the protein’s family does at a **whole-sequence level**.
>
> **The true inferential power emerges when the LLM synthesizes these orthogonal streams of information. Information at both function-module level and whole-sequence level can be cross-referenced. For a novel protein, this creates an opportunity for robust hypothesis generation.**
>
> For example, for a newly sequenced protein, InterProScan might identify a “P-loop containing nucleoside triphosphate hydrolase” domain (a feature of ATP- or GTP-binding proteins), while BLASTp identifies distant homologs that are known DNA helicases. The LLM’s task is not to retrieve an answer, but to synthesize these facts into a coherent hypothesis:
> “This protein is likely a novel DNA helicase, as its intrinsic domain is consistent with the ATP-hydrolyzing function characteristic of its evolutionary relatives.”
> This process of cross-validating information from different biological levels is a powerful engine for discovery, not a circular retrieval loop.
>
> **Q2** Could tool-calling agents solve this rather than building directly into the LLM? What are the tradeoffs?
>
> **A2:** Our context-generation pipeline can be viewed as a hard-coded, optimized agentic workflow that executes a specific, validated sequence of tools for a well-defined and critical task. A general, LLM-driven tool-calling agent represents a different point in the design space.
>
> Our context-driven method is empirically validated (Appendix E) to be robust and effective for its core task. Therefore, we do not see our method as an alternative to agent-based systems, but rather as a foundational blueprint for them. We demonstrate that for reliable biological reasoning, the primary goal should be to use tools to generate rich context first. A more advanced agent might build upon our workflow, perhaps by first running our validated pipeline and then deciding if additional tools are needed to refine the answer.
>
> **Q3:** The basis of the manuscript suggests that there is in fact valuable information encoded within the evolutionary language through sequence tokens, yet the results suggest the opposite – and that human context exclusively drives the value. Does the core contribution put a focus on the LLMs, or are you simply demonstrating that traditional bioinformatics pipelines already solve most of the problems around understanding protein function?
>
> **A3:** We appreciate this observation, as it allows us to refine a subtle but central point of our argument. We absolutely agree that the raw sequence contains a profound depth of biological and evolutionary information. Our argument is not that this information is absent, but that **current LLM architectures, due to the tokenization dilemma, are fundamentally ill-equipped to extract it directly.** Our findings do not suggest that “human context exclusively drives the value.” Rather, they show that the value from the sequence is unlocked only when it is translated into a representation that the LLM can process.
>
> Therefore, our core contribution is demonstrating that this two-step “translate-then-reason” process is vastly more effective than asking a single model to be natively fluent in both languages at once. The value is still derived from the sequence, but it is mediated through expert tools.

---

> > ### Author Response · Authors · 2025-11-21
> >
> > **Q4:** Have these results been validated against human expert annotators?
> >
> > **A4:** The ground truth for our benchmark dataset is derived directly from Swiss-Prot (UniProtKB), which is widely considered the gold standard in protein annotation. Swiss-Prot is meticulously maintained by expert biologists who manually curate each entry. Therefore, the “correct answers” in our benchmark represent a consensus of human domain experts.
> >
> > Beyond this expert-curated supervision, we further performed independent validation using wet-lab experiments. Specifically, we curated a set of novel protein sequences produced through ongoing experimental projects. These sequences were unpublished at the time of our evaluation and absent from all major reference databases, including Swiss-Prot, thereby providing a stringent, real-world test of whether the model can reason about truly unseen biology.
> >
> > We formulated the task as a binary classification problem across two biologically distinct families: **Rhodopsins** and **PETases**. For each experimentally derived sequence, the LLM was prompted to infer its family membership based solely on the provided context. As shown in Figure 6 of the revised manuscript, our context-driven method achieved 100% accuracy on Rhodopsin sequences and 97.3% accuracy on PETase, demonstrating strong concordance with expert laboratory characterization.

---

### Official Review · Reviewer_oqHE · 2025-11-01

**Soundness:** 3
**Presentation:** 3
**Contribution:** 2
**Rating:** 6
**Confidence:** 3

**Summary:**

This paper tackles how Sci-LLMs handle biomolecular sequences. It argues current methods are stuck in a "tokenization dilemma": either they treat sequences as language, breaking up important motifs, or as a separate modality, which creates an alignment gap. The authors propose a "context-driven" approach, skipping raw sequences entirely. Instead, they use bioinformatics tools (BLAST, Pfam) to create a text summary for the LLM . Their experiments show this context-only method works best, and that adding the raw sequence back in actually hurts performance, acting like noise.

**Strengths:**

The paper's "tokenization dilemma" concept is a really clear and smart way to frame a major hurdle for Sci-LLMs. The main idea—that feeding LLMs text context from tools like BLAST is better than giving them the raw sequence—is surprising but backed up well by the experiments. The finding that raw sequences just add "noise" and make things worse is a big deal. The visualizations (like in Figure 3) showing how alignment fails are also very convincing . This work is important because it questions the push for end-to-end models and offers a practical, hybrid alternative.

**Weaknesses:**

The main drawback, which the authors rightly point out, is that this method can't handle mutation effect prediction. The bio-tools (BLAST, etc.) used to create the context just aren't sensitive to tiny, single-point changes, so the context for a normal protein and its mutant look the same . This is a major limitation, as it rules out a big area of computational biology. Also, the claims about it working on DNA are mostly tucked away in the appendix, not fully explored in the main paper.

**Questions:**

Given the issue with mutations, do you have ideas for how this context-driven method could be adapted for those tasks? Maybe by using different tools that are sensitive to mutations to generate the context?

You mention your method is efficient because it avoids retraining, but running tools like InterProScan and BLAST for every query isn't free. How does the real-world inference time/cost of your pipeline compare to running a big, end-to-end model?

---

> ### Author Response · Authors · 2025-11-21
>
> Dear Reviewer oqHE,
>
> We extend our sincere gratitude for your thorough review and insightful feedback on our manuscript. We are greatly encouraged that you found the "tokenization dilemma" to be a clear and compelling framework and that you recognized the significance of our central findings, particularly the counter-intuitive result that raw sequences can act as informational noise. We address your questions below:
>
> **Q1** Given the issue with mutations, do you have ideas for how this context-driven method could be adapted for those tasks? Maybe by using different tools that are sensitive to mutations to generate the context?
>
> **A1** We thank the reviewer for this critical observation. Our framework, in its current implementation, is indeed unsuited for predicting the functional consequences of subtle mutations. We intentionally highlighted this in the limitations section (Appendix J) to be transparent about the boundaries of our current study.
>
> **However, we argue that this is not a fundamental flaw of the context-driven paradigm itself**, but rather a limitation of the specific tools chosen for the general function prediction tasks in this paper. The core principle of our paradigm—providing high-level, structured context—is flexible. To address mutation effects, we can augment or replace the context-generation tools with those sensitive to fine-grained changes. For instance:
>
> 1. Structure-Aware Homology Search: Instead of BLASTp, we could use tools like Foldseek to find structural homologs. As discussed internally, we plan to test if Foldseek yields different homologous sets for wild-type vs. mutant proteins. If so, the resulting differences in GO term annotations could form a mutation-sensitive context.
>
> 2. Specialized Predictive Tools: The output of dedicated mutation effect predictors such as SIFT and PolyPhen-2 could be directly incorporated into the context, allowing the LLM to reason about the predicted consequences of a mutation.
>
> The paradigm unlocks complex biological reasoning by flexibly integrating the most relevant expert tools. The LLM is central to this process, as it excels at the sophisticated, multi-modal evidence integration required to synthesize these diverse sources.
>
> **Q2** How does the real-world inference time/cost of your pipeline compare to running a big, end-to-end model?
>
> **A2** Thank you for raising this critical point. Your question prompted us to conduct a more detailed analysis that quantifies the trade-offs between computational cost and performance. We compare:
>
> 1. **A general LLM baseline**, feeding the raw sequence directly to the Deepseek-v3 API, which yields a performance score of **40.77**.
> 2. **A large end-to-end Sci-LLM Evolla**, requiring a high-end GPU, which yields a performance score of **59.93**.
> 3. **Our context-driven method**, using bioinformatics tools on CPU + Deepseek-v3 API, which yields a performance score of **84.99**.
>
> **Table: Comparative Analysis of Inference Cost, Time, and Performance. Cost estimates are based on AWS on-demand pricing and public API costs.**
> | Method | Mode | Input to LLM | Avg. Time (sec/seq) | Avg. Cost (USD/seq) |
> | :--- | :--- | :--- | :--- | :--- |
> | DeepSeek-V3| Single | Raw Sequence | ~30s | ~$0.0005 |
> | Evolla | Single | Raw Sequence | ~90s | ~$0.0690     |
> | Our Method | Single | Context | ~70s | ~$0.0030      |
> | Evolla | Batch | Raw Sequence | ~20s | ~$0.0152    |
> | Our Method | Batch | Context | ~0.13s | ~$0.0005     |
>
> **1. Single-Sequence Inference:**
>
> For individual queries, our method is not only dramatically more effective but also approximately **23 times cheaper and 1.3 times faster** than the specialized end-to-end model Evolla.
>
> **2. Large-Scale Batch Processing:**
>
> The true efficiency of our pipeline is most evident in high-throughput research. We modeled the cost and time to process a large dataset of 1.12 million sequences. In this realistic, large-scale scenario, our method is **nearly 30 times cheaper and 154 times faster**, representing a monumental advantage for research productivity.

---

> > ### Author Response · Authors · 2025-11-21
> >
> > To ensure transparency, we provide the basis for our cost estimations, using publicly available on-demand pricing and API costs.
> >
> > **Settings:**
> > *   **CPU Instance (Single):** An economical instance (e.g., `c6a.xlarge`) at **$0.0153 / hour**.
> > *   **CPU Instance (Batch):** Two powerful instances like c6a.24xlarge at **$2.7(each is $1.35) / hour**.
> > *   **GPU Instance (Single A100):** A single A100 instance at **$2.75 / hour**.
> > *   **LLM API Cost:** A single call to DeepSeek-V3 is estimated at **$0.000446**.
> >
> > **Single-Sequence Inference Calculations:**
> > *   **Our Method Cost: ~$0.0030**
> >     1.  **CPU Cost:** `($0.153 / 3600 seconds) * 60 seconds ≈ $0.00255`.
> >     2.  **API Cost:** `$0.000446`.
> >     3.  **Total:** `$0.00255 + $0.000446 = $0.002996 ≈ $0.0030`.
> > *   **Evolla Cost: ~$0.0690**
> >     1.  **GPU Cost:** `($2.75 / 3600 seconds) * 90 seconds ≈ $0.06875 ≈ $0.0690`.
> >
> > **Batch Processing Inference Calculations (Per-Sequence Averages):**
> > *The per-sequence averages for batch processing were derived by modeling a large-scale run to capture amortization and throughput effects.*
> > *   **Our Method Cost: ~$0.00054 per sequence**
> >     1.  **Throughput Model:** 2 powerful CPU machines process 1.12M sequences in 40 hours.
> >     2.  **CPU Cost:** `(2 machines * 40 hours * $1.35/hour) / 1.12M seq ≈ $0.000096/seq`.
> >     3.  **API Cost:** `$0.000446/seq`.
> >     4.  **Total:** `$0.000096 + $0.000446 = $0.00054/seq`.
> > *   **Evolla Cost: ~$0.0152 per sequence**
> >     1.  **Throughput Model:** 1 A100 machine processes ~180 sequences per hour.
> >     2.  **GPU Cost:** `(1 hour * $2.75/hour) / 180 seq ≈ $0.0152/seq`.

---

### Author Response · Authors · 2025-11-21

We thank all reviewers for their thoughtful evaluations and for highlighting several key strengths of our work. The reviewers agreed that **the paper introduces a clear and compelling conceptual framing through the proposed “tokenization dilemma,” which articulates a fundamental bottleneck in Sci-LLMs** (*Reviewer oqHE, 2DYC*). They also noted that **the context-only paradigm leads to surprisingly strong empirical performance**, and that the finding that raw sequences act as informational noise is both striking and well-supported across diverse models and tasks (*Reviewer oqHE, 2DYC, J4os*). In addition, the reviewers emphasized that the workb **offers a meaningful paradigm shift by positioning Sci-LLMs as reasoning engines** over structured expert knowledge rather than raw sequence decoders, with practical and generalizable benefits (*Reviewer CSFM*). The breadth and rigor of the empirical evaluation were recognized as further strengthening the contribution (*Reviewer 2DYC*). Finally, the mechanistic insights provided by our representation and layer-wise analyses, such as the demonstration of semantic misalignment and mutation-signal erosion, were highlighted as particularly informative (*Reviewer oqHE, 2DYC*).

Across the their reviews, several cross-cutting questions emerged regarding (i) computational efficiency and real-world cost, (ii) generalization to real-world unseen proteins, and (iii) reproducibility. In the revised manuscript, we have substantially strengthened the paper by addressing these concerns with new analyses, new datasets, and clearer discussions.

## **Efficiency and Cost Analysis**

Reviewers oqHE and 2DYC asked about the inference-time overhead introduced by running BLASTp and InterProScan. We conducted a comprehensive cost and latency study comparing:

(a) A general LLM baseline, feeding the raw sequence directly to the Deepseek-v3 API, which yields a performance score of 40.77.

(b) A large end-to-end Sci-LLM Evolla, requiring a high-end GPU, which yields a performance score of 59.93.

(c) Our context-driven method, using bioinformatics tools on CPU + Deepseek-v3 API, which yields a performance score of 84.99.

**Our method is ~23× cheaper and ~1.3x faster for single-sequence inference and ~30× cheaper and ~154× faster in large-scale batch scenarios than Evolla**, while also achieving substantially higher accuracy. These results are now clearly summarized in the main text (**Section 5.6**) with full details in **Appendix M**.

## **Real-World Generalization Using Unpublished Wet-Lab Sequences**

Reviewers asked whether the strong performance could reflect memorization of Swiss-Prot annotations. To rule this out, we introduced a new evaluation on unpublished wet-lab protein sequences that do not appear in Swiss-Prot or any public dataset. These sequences come from ongoing experimental projects and thus represent a true real-world setting with no possibility of memorization.

Our context-driven method (DeepSeek-V3 + context) achieves 100% accuracy on Rhodopsins and 97.3% accuracy on PETase sequences, demonstrating strong generalization to genuinely novel biological data. This new experiment is described in **Section 5.7**, with full protocol and data in **Appendix N**.

## **Reproducibility**
In response to the request for reproducibility, we now include core source code in the supplementary materials, and commit to releasing the full codebase and datasets on GitHub and Hugging Face upon acceptance.

We thank all reviewers again for their detailed feedback, which has substantially improved the clarity, rigor, and impact of the paper. We believe the revisions directly address all raised concerns and further strengthen both the conceptual and empirical contributions of the work.

---

### Public Comment · ~Xibin_Bayes_Zhou1 · 2025-11-27
**Critical Concerns: Data Leakage, Baseline Misrepresentation, and Use of Unreleased Data**

Dear reviewers and authors,

We have carefully reviewed the paper "Lost in Tokenization" and identified several critical methodological flaws and factual errors that severely undermine the validity of the reported results and conclusions. We believe these issues must be addressed publicly.

# Data Leakage in Context Generation

The paper's core claim is that a "Context-Driven" approach outperforms sequence-based models. However, the methodology described introduces data leakage.
1. The benchmark datasets are derived from **Swiss-Prot**.
2. The context generation pipeline relies on tools like **BLASTp** and **InterProScan**, which query biological databases (highly likely including Swiss-Prot itself).
3. **Crucially, regarding InterProScan**: This tool relies on member databases (e.g., Pfam) that construct HMM profiles based on large sequence alignments. Since the test set is drawn from Swiss-Prot, there is a very high probability that **the query proteins themselves were used to construct these HMM profiles**. Consequently, the system is effectively matching against signatures derived directly from the test data, rather than reasoning over external knowledge.
4. **Consequence**: The method essentially retrieves the ground truth annotations (encoded in the HMM profiles or BLAST databases) to answer questions about the same entries. Comparing this retrieval-based setup against models like Evolla (which are evaluating generalization on unseen data) is fundamentally unfair and invalidates the results in Section 5.2 and Table 1.

# Invalid Data Splits for Training-Free Methods
The authors utilize the "Hard" subset from the Evolla dataset, defined by <30% sequence identity to Evolla's training set.
1. This definition of difficulty applies **only to the Evolla model**.
2. Since the authors' approach is training-free and utilizes external databases, the distinction between "Easy" and "Hard" (relative to Evolla's training data) is irrelevant.
3. Given the data leakage identified mentioned above, the retrieval system likely has **direct access** to the ground truth for these test sequences. Consequently, these samples are trivial for the proposed method to solve via lookup, **regardless of their distance to Evolla's training set**. This explains the "**robustness**" observed in Figure 4—the consistent performance is a result of database coverage (leakage), not a sign of superior generalization.

# Misrepresentation and Misconfiguration of the Evolla Baseline
The comparison with Evolla appears to be flawed in two major aspects:
1. **Missing Modality**: Evolla utilizes **SaProt** as a protein encoder, which is designed to process structure-sequence co-evolutionary information. The authors explicitly state they used a "Sequence-Only" mode. By stripping the structural input required by the foundation model, the baseline is significantly handicapped.
2. **Misused Architecture**: In Section 5.3 and Figure 3, the authors perform a layer-wise analysis of "Evolla's Q-Former". Evolla **does not utilize a Q-Former module in its architecture**. This suggests a fundamental misunderstanding of the baseline model.
3. **Model Selection**: The authors use Evolla-10B in the paper, which is not the state-of-the-art version of the model, further weakening the baseline.

# Unauthorized Use of Unreleased Data
The paper claims to utilize the "Original Evolla Evaluation Dataset" and the "EC Number Dataset" from the Evolla study (Appendix B.1).

To our knowledge, these specific datasets **have never been publicly released** by the Evolla team, **nor has permission been granted** for their use. Consequently, the inclusion of this data implies the use of proprietary information without clear authorization. Utilizing unreleased data without explicit permission from the originating team raises concerns regarding academic integrity and adherence to standard data usage protocols.

In conclusion, we find the paper’s reported results invalid due to **data leakage**, as the proposed method effectively retrieves ground truth from external databases rather than demonstrating generalization. The study is further compromised by **a flawed baseline comparison**—which handicaps and misrepresents the Evolla architecture—and the **unauthorized use of unreleased proprietary data**, raising serious concerns regarding both scientific validity and academic integrity.

---

> ### Author Response · Authors · 2025-11-28
> **Response to Public Comment by Evolla Authors**
>
> We thank the authors of Evolla for their detailed comments. We value the opportunity to clarify our methodology and address the concerns raised regarding data usage, baseline configuration, and experimental validity.
>
> 1\. On Data Leakage (InterProScan / BLAST)
> We respectfully disagree with the characterization of our context generation pipeline as data leakage. Our method simulates a realistic inference scenario for analyzing novel proteins:
>
> * **InterProScan:** This tool performs **profile-based domain detection** using Hidden Markov Models (e.g., from Pfam). It identifies functional motifs by recognizing statistical patterns and intrinsic sequence features, **not by retrieving the query protein’s ground-truth label from a database**. It functions independently of whether the specific query sequence has been previously indexed in Swiss-Prot.
> * **BLAST:** In our pipeline, we strictly **exclude the query protein itself** from the search results. We only retrieve homologs. Transferring annotations from homologs (homology-based inference) is a foundational principle of bioinformatics and does not constitute data leakage. It represents a valid reasoning path: inferring function based on similarity to known entities, which is distinct from retrieving the ground truth label of the query itself.
>
> 2\. On Data Splits and Generalization
> Regarding the criticism that the "Hard" split is irrelevant for our training-free method:
>
> * The "Hard" split (low sequence identity to the knowledge base) serves as a critical stress test for the robustness of homology-based retrieval and reasoning. It is highly relevant for assessing how well context-driven methods perform when high-identity homologs are unavailable.
> * Furthermore, we did not limit our evaluation to this split. We demonstrated the method's superiority across diverse settings, including **established public benchmarks** like Mol-Instruction, and a **truly unseen, unpublished wet-lab dataset** of sequences completely absent from Swiss-Prot. The consistent performance across these diverse benchmarks demonstrates that our results are driven by robust context reasoning, not database leakage.
>
> 3\. On Evolla Baseline Configuration
> There appears to be a misunderstanding regarding our configuration of the Evolla baseline:
>
> * **Structural Input:** We did **not** degrade Evolla to a "Sequence-Only" mode. We faithfully reproduced the structural pipeline described in the Evolla paper. We generated structures (using PDB/AlphaFold) and encoded them using **FoldSeek** to provide structural inputs, ensuring the baseline had access to the full Structure+Sequence modality.
> * **"Q-Former" Terminology:** We acknowledge that the Evolla paper names the module responsible for compressing protein features via learnable queries the **"Sequence Compressor"** (while the subsequent injection module is the Sequence Aligner). We used the term "Q-Former" colloquially to refer to this compression component because its architecture (using learnable queries for feature extraction) is functionally analogous to the Q-Former in vision-language models. We will revise our terminology to "Evolla's Sequence Compressor" to be precise.
> * **Model Selection：** To ensure the validity of our results, we adopted the best available approach—using Evolla’s open-source maximum weight model and its official inference script—for evaluation.
>
> 4\. Clarification on Dataset Usage and "Unauthorized Data"
> We explicitly state that we did not access or use any proprietary or unreleased data from the Evolla team.
> The datasets referred to as the "Original Evolla Evaluation Dataset" and "EC Number Dataset" in our paper were **independently reconstructed** by our team strictly following the data construction protocols described in the Evolla paper. Specifically:
>
> * We retrieved raw protein entries from public databases (Swiss-Prot).
> * We reproduced the "Hard" split by filtering sequences based on the $\<30\\%$ sequence identity threshold defined in the Evolla methodology.
> * We constructed the QA pairs ourselves using the prompt templates and logic outlined in the Evolla paper.
>
> We used the term "Original" solely to distinguish this **reproduction of the Evolla benchmark** (which mimics Evolla's original QA style) from our own proposed benchmark. It is important to note that while our own benchmark utilizes the same "Hard" split sequences to ensure comparability, it employs a distinct, rigorously reconstructed QA generation method (as detailed in our paper). This terminology was used for clarity in experimental comparison, not to imply the use of unauthorized files.
>
>
> We hope these clarifications resolve the concerns regarding scientific validity and academic integrity. We are committed to rigorous comparison and believe these results highlight the genuine potential of context-driven reasoning in Sci-LLMs.

---

### Meta-Review · Area_Chair_ZYym · 2026-01-08

**Summary:**

This paper studies what inputs should be used for scientific LMs. Instead of directly using raw biological sequences such as genomic sequences, this paper found that, surprisingly, using outputs from bioinformatics tools outperforms using raw sequences or a combination of raw sequences and such outputs. Based on this result, this paper recommends treating scientific LMs as reasoning engines over expert knowledge instead of raw sequence decoders. Overall, all four reviewers are on the positive side, and they converge on 1) the finding of this paper is surprising, and 2) the "tokenization dilemma" framing, which means that the tokenization process is too granular, is clear.

There is also a public comment from Evolla authors raising several concerns: 1) potential data leakage as some of the bioinformatics tools might be trained on datasets containing benchmark data; 2) irrelevant of the Evolla hard split due to the training free method; 3) misrepresentation of Evolla; and 4) use of unreleased Evolla datasets without permission. To these points, the authors responded that: 1) they tried to minimize that by e.g. excluding the query sequence itself; 3) they'll rename Q-former to sequence compressor; 4) they reconstructed Evolla datasets.

My overall recommendation is to accept this paper, despite the above concerns assuming good faith. But I request the authors to make the following clarifications in their camera ready version to address the above concerns: 1) clarify the dependence on Swiss-Prot-derived tools; 2) clarify the Evolla baseline configuration and soften any over-strong SOTA claims; and 3) clarify dataset name as reconstruction.

**Reviewer Concerns:**

1. memorization / data leakage due to data potentially used for pretraining LMs. This is partly mitigated by using unpublished wet-lab sequences.
2. computational cost due to using those bioinformatics tools. This is partly addressed by measuring the latency of different methods.
3. circularity in using expert annotations (relying on curated annotations to predict curated annotations). This is clarified by the intended use of the system.

**Reviewer Scores:**

I don't see significant misunderstanding from the reviewers' part, and they overall has a very small variance in scores. Therefore, I don't think they would have changed their scores.

---

### Decision · Program_Chairs · 2026-01-26

Accept (Poster)